# CE$^4$L: Continual Ego, Exo, and Ego-Exo Learning

**Hongwei Yan** [1 2]  **Kanglei Zhou** [3]  **Yuchen Liu** [4]  **Qingyu Shi** [5]  **Yi Zhong** [1 2]  **Liyuan Wang** [3]

## Abstract

Perception for embodied agents is video-based, often multi-view (ego, exo, or both), and inherently continual, with simultaneous task and viewpoint shifts. Yet continual learning (CL) remains dominated by exo-only recognition tasks, obscuring behavior under these real-world coupled shifts. We introduce **C**ontinual **E**go, **E**xo, and **E**go-**E**xo **L**earning (**CE$^4$L**), a unified multi-view CL benchmark spanning four representative tasks: cross-view referenced skill assessment, temporal action segmentation, cross-view association, and action anticipation & planning. CE$^4$L highlights challenges largely absent in prior CL benchmarks, including cross-view correspondence, view-dependent asynchrony, and heterogeneous semantic objectives. To this end, we propose **V**ideo **I**ncremental **S**ubspace-routed **T**ask **A**dapters (**VISTA**), a parameter-efficient baseline method that stores task-specific updates in lightweight adapters and performs training-free routing via residual distance to task-specific whitened subspaces estimated from second-order statistics. Extensive experiments demonstrate the significantly varied efficacy of representative CL methods across CE$^4$L settings, while VISTA is consistently competitive and achieves state-of-the-art overall performance. Our source code for benchmarks and methods is available at CE$^4$L.

## 1. Introduction

The pursuit of embodied intelligence is gradually approaching real-world scenarios that mirror human life, requiring agents to learn continually from incoming experiences, adapt to new environments, acquire skills by observing expert demonstrations, and sustain mastered capabilities (Kim et al., 2024; Cheang et al., 2024; Black et al., 2026; Intelligence et al., 2025; Liu et al., 2025). However, the continual learning (CL) process faces a unique challenge in the embodied context: unlike the well-distributed and ideally static datasets typical of standard supervised training, the sensory signals received by an embodied agent, including visual stimuli, are temporally correlated and highly dependent on its physical viewpoint, a characteristic central to egocentric vision (Punamiya et al., 2025; Yang et al., 2025). While recent large-scale pretraining has driven remarkable progress in understanding such egocentric videos (Pramanick et al., 2023; Zhao et al., 2023; Chen et al., 2022), existing CL protocols have not kept pace for this domain. They often abstract away two properties that define embodied experience: strong viewpoint bias and heterogeneous semantic objectives (Park et al., 2021; Villa et al., 2022; Maraghi & Faez, 2022; Pian et al., 2023).

Bridging this gap is not a straightforward matter of scaling existing CL paradigms from 2D images to video, where viewpoint bias fundamentally changes what should be preserved across time (Xu et al., 2023; Grauman et al., 2024; Plizzari et al., 2024). A learner can maintain performance on previously seen tasks within the training view while losing the cross-view invariance that makes its representation transferable, or conversely it can appear to forget simply because evaluation shifts to a mismatched view. This entanglement makes it difficult to interpret CL progress using single-view and single-task paradigms alone, and it obscures which algorithmic ingredients genuinely improve continual video learning rather than overfitting to the current view.

To address these challenges, we introduce **C**ontinual **E**go, **E**xo and **E**go-**E**xo **L**earning (**CE$^4$L**), a comprehensive benchmark suite designed to facilitate CL in the context of multi-view video understanding. Built upon the EgoExoLearn (Huang et al., 2024) video dataset, CE$^4$L establishes a rigorous evaluation framework that explicitly disentangles within-view retention from cross-view generalization.

---

[1]School of Life Sciences, IDG/McGovern Institute for Brain Research, Tsinghua University, Beijing, China [2]Tsinghua-Peking Center for Life Sciences [3]Department of Psychological and Cognitive Sciences, Tsinghua University, Beijing, China [4]Department of Computer Science, Tsinghua University, Beijing, China [5]Institute for Interdisciplinary Information Sciences, Tsinghua University, Beijing, China. Correspondence to: Liyuan Wang <wly19@tsinghua.org.cn>.

*Proceedings of the 43$^{rd}$ International Conference on Machine Learning*, Seoul, South Korea. PMLR 306, 2026. Copyright 2026 by the author(s).

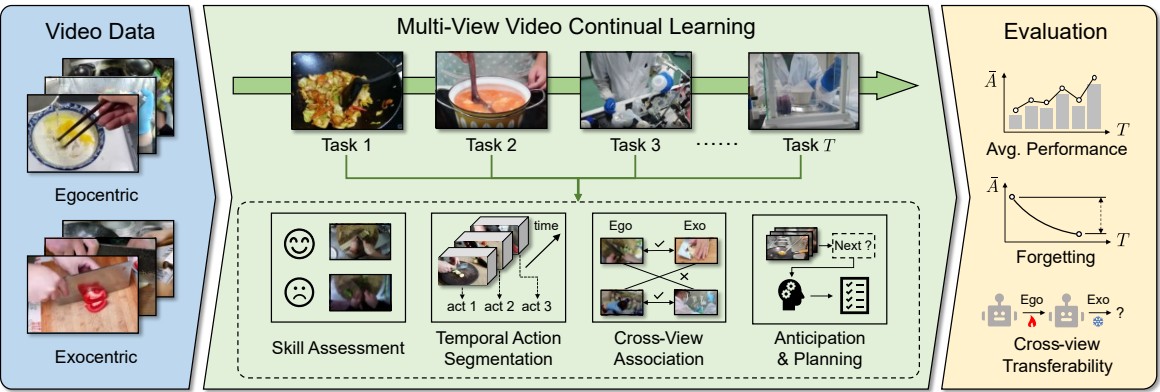

*Figure 1.* **CE$^4$L benchmark overview.** By constructing continuous task stream for both ego- and exocentric videos, CE$^4$L comprehensively evaluates the within-view performance and cross-view transferability of continual learners through four diverse benchmarks.

We instantiate four complementary problems as a sequence of tasks: **continual pairwise skill assessment**, **temporal action segmentation**, **cross-view association**, and **action anticipation & planning**. Crucially, our protocol supports both single-view and multi-view training regimes, enabling a grounded assessment of a model's cross-view transferability as well as its plasticity to viewpoint variations.

Under CE$^4$L, we perform a systematic study of representative continual paradigms and reveal a critical insight: **retaining past performance is often hindered by the learner drifting toward view-specific shortcuts that fail to transfer**. Concretely, the relative efficacy of replay, regularization, and ensemble-based methods vary substantially across objectives (Tables 2–5), and in several scenarios cross-view evaluation exposes viewpoint-conditioned mismatch that can persist even when within-view forgetting is reduced (Figs. 6–8). These findings motivate evaluation protocols that go beyond average accuracy on single-view recognition and make cross-view transfer explicit.

Motivated by these findings, we propose **V**ideo **I**ncremental **S**ubspace-routed **T**ask **A**dapters (**VISTA**), a parameter-efficient continual learner that serves as a strong replay-free baseline for continual multi-view video learning. VISTA stores task knowledge in lightweight adapters and performs training-free inference by routing inputs to a small adapter ensemble via residual distances to task-specific feature subspaces. Across CE$^4$L, VISTA achieves the strongest overall performance among CL baselines while remaining efficient in memory and compute (Tables 2–5; Table 10).

We summarize our contributions as follows:

**I.** We introduce CE$^4$L, a unified benchmark that evaluates CL across heterogeneous video tasks (ego, exo, and mixed-view), providing protocols that separate forgetting from cross-view transfer capability.

**II.** We propose VISTA, an efficient adapter-based method that utilizes task subspace geometry for robust, training-free task inference in high-dimensional video representations.

**III.** We provide a comprehensive evaluation of representative CL baselines under the CE$^4$L protocols and diagnose failure modes related to view-conditioned interference, opening up a new avenue for embodied CL.

## 2. Related Work

**Egocentric and ego-exo video learning.** Egocentric video understanding has progressed from curated first-person datasets to large-scale, multi-view procedural corpora that connect first-person observations with third-person demonstrations (Damen et al., 2022; Liu et al., 2022; Huang et al., 2024; Grauman et al., 2024). Existing benchmarks and representation models have enabled strong offline perception, cross-view association, and skill understanding (Pramanick et al., 2023; Zhao et al., 2023; Chen et al., 2022), but they rarely evaluate how such capabilities are retained when tasks and viewpoints evolve continually.

**Video continual learning.** Video continual learning extends continual learning beyond static images by introducing temporal dynamics, high-dimensional inputs, and video-specific forgetting patterns (Park et al., 2021; Villa et al., 2022; Wang et al., 2024a). Prior work has explored replay, regularization, prompt tuning, and video-language continual adaptation (Pei et al., 2023; Villa et al., 2023; Tang et al., 2024; Tan et al., 2025), yet most protocols remain centered on single-view recognition rather than the coupled viewpoint and objective shifts faced by embodied agents.

**Pretraining-based continual learning.** Recent continual learning methods increasingly adapt frozen pretrained models with lightweight prompts or adapters, often relying on task inference or routing to select task-specific components (Wang et al., 2022d;c;a; 2023b). In this work, we follow this parameter-efficient direction, but use training-free subspace routing for multi-view video streams where a learned router may itself be vulnerable to non-stationarity (McDonnell et al., 2024; Zhuang et al., 2022). A more detailed literature review is provided in Appendix A.

*Table 1.* **CE$^4$L benchmark configurations.** Details of each multi-view video continual learning problem constructed in CE$^4$L.

| CE$^4$L Scenario Info. | Skill Assessment | Action Segmentation | Cross-View Association | Action Anticipation | Action Planning |
|---|---|---|---|---|---|
| Number of Tasks | 4 | 4 | 5 | 8 | 4 |
| Task Division | Action group | Procedural task | Procedural task | Procedural task | Procedural task |
| Backbone Network | I3D+RAAN | I3D+MS-TCN | TimeSformer-B + CLIP-text | CLIP+TA3N | CLIP+TA3N |
| Evaluation Metrics | Ranking Acc ($\uparrow$) | Acc/Edit/F1 ($\uparrow$) | Top-1 Acc ($\uparrow$) | Top-5 Recall ($\uparrow$) | ED@8 ($\downarrow$) |

## 3. CE$^4$L Benchmark

### 3.1. General Setup

Traditional CL benchmarks focus primarily on exocentric recognition with limited distribution shifts. This abstraction, however, fails under structured viewpoint changes inherent to embodied agents, where underlying procedure can appear substantially different across ego and exo perspectives. A learner may preserve within-view performance yet lose transferable cross-view knowledge. CE$^4$L explicitly addresses this by coupling task-incremental streams with evaluation metrics that separately measure **within-view retention**, zero-shot **cross-view generalization**, and a **co-training baseline regime** that provides both views to distinguish stability from the lack of multi-view evidence.

CE$^4$L comprises four CL benchmarks that emphasize complementary challenges in cross-perspective video learning (see Fig. 1): **cross-view referenced skill assessment**, **temporal action segmentation**, **cross-view association**, and **action anticipation & planning**. These four benchmarks are built upon EgoExoLearn (Huang et al., 2024), which is collected in an ego-exo demonstration-following setting and covers eight high-level procedural tasks spanning both daily kitchens (five tasks) and professional laboratories (three tasks). The proposed CE$^4$L involves two kinds of task streams: (**I**) tasks with full procedures leverage the inherent eight-task partition; and (**II**) tasks with trimmed clips are instead split by grouping action categories into disjoint subsets. Detailed configurations are summarized in Table 1.

Formally, we denote a pretrained video encoder as $f_\theta$, and a task head as $g_\psi$, yielding a predictor $g_\psi \circ f_\theta$. A continual data stream comprises a sequence of task datasets $\{\mathcal{D}_t\}_{t=1}^{T}$ presented in order[1]. After training on task $t$, we evaluate on the union of seen tasks and record the performance matrix $A$, where $A_{t,j}$ denotes performance on task $j \leq t$ after learning up to $t$. We record $\{A_{t,j}\}_{j \leq t}$, and calculate final average task performance $\bar{A}_T$ and average task forgetting $\bar{F}_T$ for all benchmarks and corresponding metrics. More details of the benchmark setup are provided in Appendix B.

---

[1]In practice, the CL stream consists of sequential sessions formed by shuffling tasks. We use "task" for narrative convenience, reserving "session" only for disambiguation.

### 3.2. Continual Skill Assessment

**Problem setup.** This task is formulated as a stream of action-specific pairwise rankings. Each training sample consists of two egocentric clips $\{v_1, v_2\}$ of the same atomic action; the model must predict which demonstrates higher skill. When an exocentric reference is available, the model compares egocentric candidates against it, requiring robustness to asynchrony and viewpoint shifts. We partition action categories into disjoint subsets to form the continual data stream, using ranking accuracy as the primary metric.

**Backbone architecture.** We adopt the ranking-based model RAAN (Doughty et al., 2019) built on pretrained I3D features (Carreira & Zisserman, 2017), which outputs a scalar score per clip (Huang et al., 2024); prediction is the score difference between the two egocentric candidates. We consider two reference-aware variants: a triplet-style objective (TL) that pulls the higher-skill egocentric embedding closer to the exocentric reference than the lower-skill one, and a relation-network variant (RN) that conditions scoring on the reference via feature interaction (Sung et al., 2018). This benchmark isolates CL behavior under pairwise supervision with an explicit cross-view reference.

### 3.3. Continual Action Segmentation

**Problem setup.** Continual temporal action segmentation evaluates dense step parsing for long procedural videos. Given a video $v$, the model predicts an action label for each frame. We construct a task stream over procedural activities and train sequentially, where each task introduces a new activity with its own action distribution and temporal structure. We report within-view performance and the cross-view/co-training regimes defined in Sec. 3.1.

**Backbone architecture.** We use MS-TCN (Farha & Gall, 2019) on per-frame I3D features (Carreira & Zisserman, 2017). Performance is measured by frame-wise accuracy, edit score, and segmental F1 at multiple overlap thresholds, summarized into task-wise aggregates for stable continual comparisons across different activity temporal resolution.

### 3.4. Continual Cross-View Association

**Problem setup.** Cross-view association requires retrieving semantically corresponding clips across ego- and exocen-

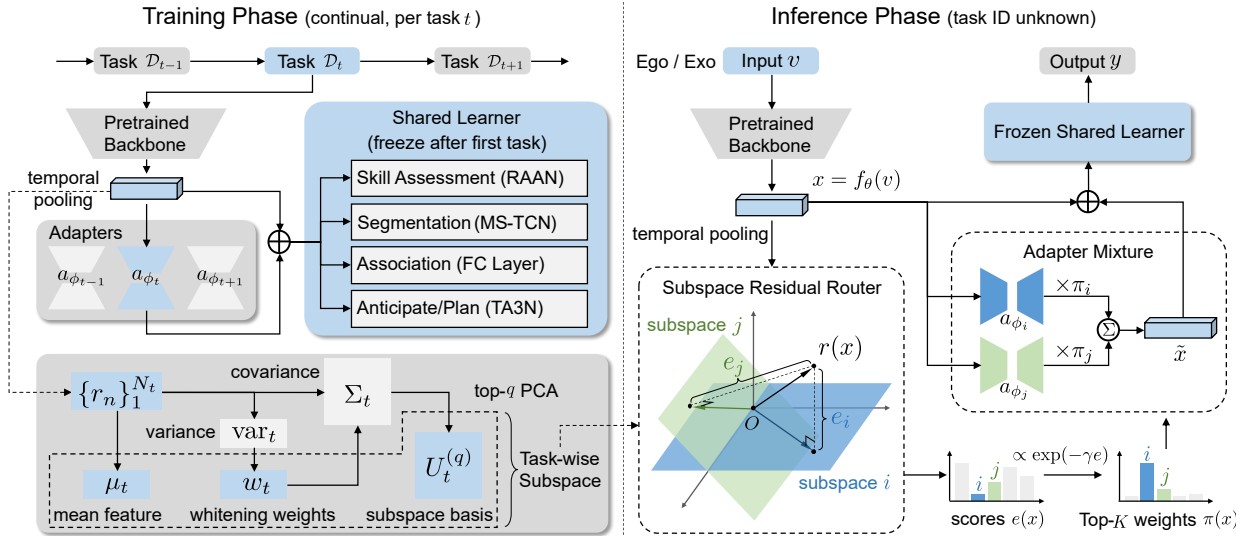

*Figure 2.* **VISTA method overview.** VISTA employs task-specific feature adapters for CL and a training-free router for inference. The router assigns mixture weights by calculating the residual distance between input features and task subspaces, defined by accumulated second order statistics. Finally, adapted features are processed by the shared learner module which is frozen after the first task.

tric videos under temporal asynchrony and large viewpoint changes. We formulate it as multiple-choice retrieval: given a query clip from one view, select its match in the other view from a fixed-size candidate set (Huang et al., 2024; Xu et al., 2021; Lin et al., 2022; Pramanick et al., 2023). We report Top-1 accuracy for both Ego→Exo and Exo→Ego directions, and form a continual data stream by grouping queries by their underlying procedural activities.

**Backbone architecture.** We use a pretrained dual-encoder video-text backbone with a time-aware visual encoder (TimeSformer (Bertasius et al., 2021)) and a CLIP text encoder (Radford et al., 2021) to obtain clip embeddings (Lin et al., 2022), trained with contrastive objectives under paired supervision. At inference, association reduces to similarity between query and candidate embeddings, directly probing whether sequential updates preserve cross-view alignment.

### 3.5. Continual Action Anticipation & Planning

**Problem setup.** CE⁴L includes both short-horizon action anticipation and long-horizon action planning (Huang et al., 2024). For anticipation, the input is a short context window and the target is the next action category (verb and noun), treated as multi-label prediction and evaluated by Top-5 recall (Damen et al., 2022). For planning, the input is also a context window but the target is a fixed-length sequence of future coarse steps, evaluated by an edit-distance style metric aggregated over horizons (Grauman et al., 2022). In both problems, we define the continual data stream over procedural activities and again implement the within-view, cross-view, and co-training regimes described in Sec. 3.

**Backbone architecture.** We adopt a lightweight temporal relation model TA3N (Chen et al., 2019) on top of a frozen CLIP encoder (Radford et al., 2021) at a fixed frame rate. The model aggregates temporal segments into a video-level representation and predicts either the next-step category (anticipation) or the future step sequence (planning).

## 4. VISTA Method

In this section, we present the proposed VISTA for multi-view video CL and give detailed description of how it adapts the pretrained representation efficiently through adapters and assembles them by utilizing the corresponding subspace geometry. See Fig. 2 for an overview.

### 4.1. Task Aware Adapters

We build VISTA on top of a pretrained video encoder $f_\theta$, which maps an input clip $v$ to a sequence of features $x = f_\theta(v) \in \mathbb{R}^{L \times C}$, where $L$ is the temporal length and $C$ is the channel dimension. In CL, directly updating $\theta$ of a large pretrained model is data intensive, and fine-tuning across a non-stationary data stream often causes severe interference, especially when the stream mixes heterogeneous tasks and viewpoints (Zhou et al., 2024a). VISTA therefore introduces a lightweight task-indexed adapter bank $\{a_{\phi_t}\}_{t=1}^T$ that modulates intermediate features before converting them to task head $g_\psi$ while keeping adaptation plasticity localized to each task in a stream of $T$ tasks (Wang et al., 2025d; Li et al., 2025; Wang et al., 2025b).

Each adapter is a residual bottleneck module applied independently at every time step (Rebuffi et al., 2017). Let

$x_i \in \mathbb{R}^C$ denote the feature vector at time index $i$,

$$a_\phi(x)_i = x_i + \alpha \, W_{\text{up}} \, \sigma(W_{\text{down}} \, \text{LN}(x_i)), \qquad (1)$$

where $\text{LN}(\cdot)$ is layer normalization, $\sigma(\cdot)$ is a GELU non-linearity, $W_{\text{down}} \in \mathbb{R}^{r \times C}$ and $W_{\text{up}} \in \mathbb{R}^{C \times r}$ are trainable linear maps with bottleneck dimension $r \ll C$, and $\alpha \in \mathbb{R}$ is a learnable residual scale initialized to 0 so that $a_\phi$ starts from an identity mapping. This parameterization ensures that each newly introduced adapter can be optimized stably without disrupting the pretrained representation geometry at initialization, while still permitting task-specific specialization as training proceeds in an efficient way.

During CL, when the data stream is at task $t$, $a_{\phi_t}$ is initialized with parameter of the last one $a_{\phi_{t-1}}$, supporting cross-task knowledge transfer with a warm-start. VISTA only updates the corresponding $a_{\phi_t}$ in each task, and freeze the shared head $g_\psi$ after the first task to avoid disrupting the representation calibrated by the first adapter. At inference time, the task identity is not available. Instead, VISTA predicts a small set of candidate tasks and forms a mixture of adapters to produce an adapted feature sequence (Yu et al., 2024; 2025). Concretely, given a mixture specification $\{(t_\ell, \pi_\ell)\}_{\ell=1}^K$ with $\sum_\ell \pi_\ell = 1$ and $\pi_\ell \geq 0$, we compute

$$\tilde{x} = \sum_{\ell=1}^K \pi_\ell \, a_{\phi_{t_\ell}}(x), \qquad (2)$$

which reduces to the task-aware adapter when the router is confident ($K = 1$) and yields a soft interpolation when the input lies near task boundaries. The key remaining question is how to infer $\{(t_\ell, \pi_\ell)\}$ without labels, which is addressed by our subspace residual router below.

### 4.2. Subspace Residual Router

VISTA routes each input to a small subset of previously learned adapters by comparing its representation against a set of task-specific subspaces. The router is training-free at inference time: it only uses second-order statistics accumulated from past tasks, and does not require learning an additional classifier over task identities. This design benefits from the strong representation capabilities of pretrained models, and is crucial in CE$^4$L, where task boundaries can be subtle and view shifts can dominate appearance cues. In mixed ego-exo streams, viewpoint changes can shift both feature centers and dominant variation directions; mean or prototype routers mainly capture the former and can become biased toward view-specific shortcuts. VISTA therefore normalizes task-conditioned variance through whitening and fits residual subspaces to capture task-specific variation directions, making routing better aligned with the multi-view difficulty exposed by CE$^4$L.

**Router representation.** Given a feature sequence $x \in \mathbb{R}^{L \times C}$ produced by $f_\theta$, we first extract a compact routing vector $r(x) \in \mathbb{R}^d$. We split the temporal axis into $M$ contiguous chunks and apply mean pooling within each chunk, then concatenate the pooled vectors to reduce the dimension:

$$r(x) = \left[ \frac{1}{|I_1|} \sum_{i \in I_1} x_i \, \Big\| \cdots \Big\| \, \frac{1}{|I_M|} \sum_{i \in I_M} x_i \right] \in \mathbb{R}^d, \qquad (3)$$
$$d = MC,$$

where $\{I_m\}_{m=1}^M$ is a partition of $\{1, \ldots, L\}$ and $\|\cdots\|$ denotes concatenation. Choice of $M$ depends on the input video length to balance the trade-off between information and efficiency, and such temporal pooling process provides a stable and low-variance routing signal.

**Whitened subspace construction.** At the end of learning task $t$, we collect routing vectors $\{r_n\}_{n=1}^{N_t}$ from the training data of task $t$ and estimate their mean and diagonal variance:

$$\mu_t = \frac{1}{N_t} \sum_{n=1}^{N_t} r_n, \quad \text{var}_t = \frac{1}{N_t} \sum_{n=1}^{N_t} r_n \odot r_n - \mu_t \odot \mu_t, \quad (4)$$

where $\odot$ is the Hadamard product. We then define an element-wise whitening weight vector $w_t = \sqrt{\text{var}_t + \epsilon}$, with a small $\epsilon > 0$ for numerical stability. Using $w_t$, we compute centered whitened residuals $z_n = (r_n - \mu_t) \odot w_t$ and their covariance matrix in the whitened subspace:

$$\Sigma_t = \frac{1}{N_t - 1} \sum_{n=1}^{N_t} z_n z_n^\top. \qquad (5)$$

Let $U_t \in \mathbb{R}^{d \times q}$ denote the top-$q$ eigenvectors of $\Sigma_t$, namely, the principal directions in the subspace. In addition to these variation directions, we explicitly include another whitened mean direction to better capture the space geometry

$$m_t = \frac{\mu_t \odot w_t}{\|\mu_t \odot w_t\|_2}. \qquad (6)$$

Finally, we build an *augmented* orthonormal basis $B_t \in \mathbb{R}^{d \times (q+1)}$ by QR factorization over the columns of $[m_t \, U_t]$. This augmentation couples first-order shift (mean) and second-order variation (PCA) into a single subspace, enabling $B_t$ to serve as an effective task signature.

**Residual-based routing score.** Given a query routing vector $r$, we compute its task-conditional whitened coordinates $\tilde{r}_t = r \odot w_t$ and evaluate how well $\tilde{r}_t$ is explained by the augmented basis $B_t$. The whitened subspace residual score is defined as the complement of the explained energy ratio:

$$e_t(r) = 1 - \frac{\|B_t^\top \tilde{r}_t\|_2^2}{\|\tilde{r}_t\|_2^2 + \epsilon}. \qquad (7)$$

This score is non-negative and decreases when the query lies close to the task subspace in the whitened metric. It is also

*Table 2.* Performance of CL methods on the **continual skill assessment benchmark**. We report the final average ranking accuracy $\bar{A}_T$ (%, ↑) and forgetting metrics $\bar{F}_T$ (%, ↓).

| Method | Final Average Accuracy $\bar{A}_T$(%, ↑) | | | Final Average Forgetting $\bar{F}_T$(%, ↓) | | |
|---|---|---|---|---|---|---|
| | RAAN | +RN | +TL | RAAN | +RN | +TL |
| Joint | $80.42_{\pm0.14}$ | $80.17_{\pm0.06}$ | $80.41_{\pm0.08}$ | – – | – – | – – |
| Fine-tuning | $72.94_{\pm2.15}$ | $72.45_{\pm2.21}$ | $71.63_{\pm2.13}$ | $11.30_{\pm1.35}$ | $10.97_{\pm1.64}$ | $13.20_{\pm1.94}$ |
| ER | $77.24_{\pm0.56}$ | $77.39_{\pm0.50}$ | $78.27_{\pm0.73}$ | $3.94_{\pm2.30}$ | $3.13_{\pm2.83}$ | $2.39_{\pm0.95}$ |
| DER++ | $\underline{79.46}_{\pm0.26}$ | $78.60_{\pm0.30}$ | $\underline{79.47}_{\pm0.07}$ | $0.98_{\pm1.24}$ | $1.17_{\pm1.70}$ | $1.09_{\pm0.81}$ |
| EWC | $73.34_{\pm2.28}$ | $72.59_{\pm1.95}$ | $71.05_{\pm2.00}$ | $10.62_{\pm1.75}$ | $11.32_{\pm2.26}$ | $14.22_{\pm2.44}$ |
| LwF | $72.61_{\pm1.73}$ | $68.87_{\pm2.58}$ | $71.83_{\pm2.11}$ | $2.21_{\pm3.63}$ | $12.52_{\pm3.20}$ | $0.49_{\pm2.24}$ |
| L2P+ | $72.03_{\pm1.48}$ | $71.43_{\pm2.56}$ | $71.38_{\pm0.97}$ | $11.69_{\pm3.64}$ | $12.52_{\pm1.20}$ | $14.22_{\pm3.81}$ |
| S-Prompt+ | $79.17_{\pm0.30}$ | $\underline{77.75}_{\pm0.41}$ | $78.05_{\pm0.78}$ | $\underline{0.05}_{\pm0.03}$ | $\mathbf{0.00}_{\pm0.21}$ | $\underline{0.13}_{\pm0.17}$ |
| VISTA (ours) | $\mathbf{80.38}_{\pm0.32}$ | $\mathbf{79.58}_{\pm0.81}$ | $\mathbf{80.41}_{\pm0.48}$ | $\mathbf{-0.19}_{\pm0.51}$ | $\underline{0.19}_{\pm0.19}$ | $\mathbf{0.12}_{\pm0.14}$ |

*Table 3.* Performance of CL methods on the **continual temporal action segmentation benchmark** (test split). We report the final average frame-wise accuracy $\bar{A}_T$ (%, ↑).

| Method | Train: Ego | | Train: Exo | | Train: Ego+Exo | |
|---|---|---|---|---|---|---|
| | Eval: Ego | Eval: Exo | Eval: Ego | Eval: Exo | Eval: Ego | Eval: Exo |
| Joint | $66.12_{\pm0.26}$ | $22.66_{\pm3.83}$ | $24.51_{\pm0.92}$ | $36.98_{\pm0.66}$ | $66.41_{\pm0.66}$ | $38.26_{\pm1.12}$ |
| Fine-tuning | $36.26_{\pm0.58}$ | $10.16_{\pm1.91}$ | $13.89_{\pm2.20}$ | $20.05_{\pm1.11}$ | $35.51_{\pm1.82}$ | $20.32_{\pm1.75}$ |
| ER | $45.19_{\pm3.03}$ | $10.87_{\pm6.34}$ | $18.43_{\pm2.24}$ | $30.76_{\pm0.95}$ | $43.93_{\pm5.28}$ | $29.67_{\pm0.79}$ |
| DER++ | $43.97_{\pm4.76}$ | $10.02_{\pm2.05}$ | $18.23_{\pm5.12}$ | $30.71_{\pm2.78}$ | $45.22_{\pm3.18}$ | $30.04_{\pm3.94}$ |
| EWC | $41.35_{\pm8.73}$ | $8.15_{\pm2.77}$ | $\underline{19.67}_{\pm5.05}$ | $30.88_{\pm2.44}$ | $45.11_{\pm2.85}$ | $30.35_{\pm2.58}$ |
| LwF | $44.95_{\pm3.39}$ | $\underline{12.65}_{\pm1.65}$ | $19.07_{\pm2.49}$ | $30.53_{\pm1.86}$ | $\underline{45.30}_{\pm2.83}$ | $30.35_{\pm4.53}$ |
| L2P+ | $\underline{47.46}_{\pm2.00}$ | $12.33_{\pm3.10}$ | $18.67_{\pm3.23}$ | $\mathbf{32.18}_{\pm1.57}$ | $44.88_{\pm3.08}$ | $\underline{30.84}_{\pm2.05}$ |
| S-Prompt+ | $38.69_{\pm0.97}$ | $7.52_{\pm2.10}$ | $15.38_{\pm1.74}$ | $28.01_{\pm1.03}$ | $40.85_{\pm0.70}$ | $23.72_{\pm0.55}$ |
| VISTA (ours) | $\mathbf{47.72}_{\pm0.79}$ | $\mathbf{12.93}_{\pm2.33}$ | $\mathbf{20.03}_{\pm1.16}$ | $\underline{30.90}_{\pm1.75}$ | $\mathbf{45.35}_{\pm1.34}$ | $\mathbf{31.41}_{\pm0.87}$ |

*Table 4.* Performance of CL methods on the **continual cross-view association benchmark** (test split). We report the final average Top-1 association accuracy $\bar{A}_T$ (%, ↑) for both query directions.

| Method | Train: Ego | | Train: Exo | | Train: Ego+Exo | |
|---|---|---|---|---|---|---|
| | Ego→Exo | Exo→Ego | Ego→Exo | Exo→Ego | Ego→Exo | Exo→Ego |
| Joint | $40.07_{\pm0.90}$ | $37.97_{\pm1.05}$ | $38.47_{\pm0.41}$ | $45.87_{\pm0.59}$ | $47.10_{\pm0.36}$ | $46.53_{\pm2.16}$ |
| Fine-tuning | $32.63_{\pm0.61}$ | $25.30_{\pm2.48}$ | $31.37_{\pm1.33}$ | $36.47_{\pm2.04}$ | $41.13_{\pm1.99}$ | $35.57_{\pm0.17}$ |
| ER | $34.20_{\pm1.22}$ | $27.27_{\pm1.52}$ | $\underline{32.53}_{\pm1.67}$ | $\underline{36.80}_{\pm1.71}$ | $\underline{43.47}_{\pm0.25}$ | $\mathbf{38.83}_{\pm1.46}$ |
| DER++ | $33.90_{\pm2.20}$ | $26.73_{\pm1.14}$ | $31.90_{\pm1.63}$ | $36.23_{\pm1.42}$ | $43.10_{\pm1.20}$ | $37.97_{\pm1.44}$ |
| EWC | $32.40_{\pm0.67}$ | $23.63_{\pm0.85}$ | $29.83_{\pm0.29}$ | $34.40_{\pm2.13}$ | $41.77_{\pm0.90}$ | $36.13_{\pm1.02}$ |
| LwF | $30.90_{\pm3.86}$ | $25.27_{\pm3.08}$ | $31.50_{\pm0.70}$ | $36.73_{\pm2.07}$ | $41.20_{\pm1.28}$ | $35.20_{\pm0.14}$ |
| L2P+ | $33.77_{\pm0.88}$ | $25.13_{\pm0.78}$ | $29.70_{\pm1.77}$ | $33.20_{\pm2.68}$ | $39.90_{\pm0.85}$ | $36.17_{\pm0.97}$ |
| S-Prompt+ | $\underline{34.93}_{\pm0.98}$ | $\underline{30.33}_{\pm0.80}$ | $30.78_{\pm1.42}$ | $31.26_{\pm1.24}$ | $40.57_{\pm0.33}$ | $36.60_{\pm1.15}$ |
| VISTA (ours) | $\mathbf{34.97}_{\pm1.10}$ | $\mathbf{30.37}_{\pm0.83}$ | $\mathbf{34.60}_{\pm1.89}$ | $\mathbf{37.67}_{\pm6.38}$ | $\mathbf{44.74}_{\pm1.18}$ | $\underline{38.29}_{\pm1.16}$ |

*Table 5.* Performance of CL methods on the **continual action planning benchmark**. We report the final average edit distance ED@8 (↓) over a fixed length of 8 steps.

| Method | Train: Ego | | Train: Exo | | Train: Ego+Exo | |
|---|---|---|---|---|---|---|
| | Eval: Ego | Eval: Exo | Eval: Ego | Eval: Exo | Eval: Ego | Eval: Exo |
| Joint | $82.13_{\pm0.54}$ | $84.54_{\pm0.38}$ | $84.70_{\pm0.22}$ | $76.15_{\pm0.49}$ | $81.60_{\pm0.24}$ | $76.14_{\pm0.97}$ |
| Fine-tuning | $86.69_{\pm0.53}$ | $87.91_{\pm0.34}$ | $88.33_{\pm0.36}$ | $83.23_{\pm0.45}$ | $85.53_{\pm0.56}$ | $82.68_{\pm1.21}$ |
| ER | $83.14_{\pm0.31}$ | $86.84_{\pm0.72}$ | $88.01_{\pm0.77}$ | $81.98_{\pm0.61}$ | $83.47_{\pm0.64}$ | $80.70_{\pm0.47}$ |
| DER++ | $\underline{82.64}_{\pm0.10}$ | $86.97_{\pm0.80}$ | $87.16_{\pm0.87}$ | $\underline{81.01}_{\pm0.31}$ | $\underline{82.70}_{\pm0.18}$ | $\mathbf{79.04}_{\pm1.40}$ |
| EWC | $83.81_{\pm0.61}$ | $86.03_{\pm0.43}$ | $88.67_{\pm0.39}$ | $81.78_{\pm0.62}$ | $84.60_{\pm0.37}$ | $81.88_{\pm1.17}$ |
| LwF | $\mathbf{82.42}_{\pm0.64}$ | $\underline{85.89}_{\pm0.60}$ | $86.36_{\pm1.20}$ | $82.31_{\pm0.65}$ | $\mathbf{82.03}_{\pm0.61}$ | $\underline{80.67}_{\pm1.31}$ |
| L2P+ | $83.51_{\pm1.16}$ | $86.08_{\pm0.28}$ | $86.61_{\pm1.07}$ | $81.94_{\pm0.53}$ | $83.76_{\pm1.16}$ | $82.38_{\pm3.58}$ |
| S-Prompt+ | $84.02_{\pm0.51}$ | $87.33_{\pm1.02}$ | $\underline{85.07}_{\pm0.64}$ | $82.18_{\pm0.83}$ | $85.40_{\pm0.59}$ | $81.57_{\pm0.80}$ |
| VISTA (ours) | $83.02_{\pm0.52}$ | $\mathbf{85.46}_{\pm0.77}$ | $\mathbf{84.09}_{\pm1.34}$ | $\mathbf{80.36}_{\pm0.71}$ | $82.81_{\pm0.21}$ | $80.74_{\pm0.92}$ |

scale-stable due to the normalization by $\|\tilde{r}_t\|_2^2$. When an input provides multiple correlated feature sequences (e.g., a pair of clips in skill assessment), we compute residual scores separately and average them to obtain mixture weights, which reduces variance without requiring supervision.

Let $\mathcal{S}$ be the set of tasks observed so far. We convert the residual scores into a posterior over tasks via softmax:

$$p_t(r) = \frac{\exp(-\gamma\, e_t(r))}{\sum_{j \in \mathcal{S}} \exp(-\gamma\, e_j(r))}, \qquad (8)$$

where $\gamma > 0$ controls the sharpness of routing. We then select the top-$K$ tasks with largest $p_t(r)$ and normalize their probabilities to obtain mixture weights $\{\pi_\ell\}_{\ell=1}^K$. These weights are used to mix task adapters as in the adapter mixture above, yielding a task-agnostic inference rule that interpolates between hard routing and soft sharing depending on the confidence implied by residual gaps and it also mitigates performance degradation caused by misrouting.

## 5. Experiments

### 5.1. Experimental Setup

**Dataset.** CE⁴L benchmarks are constructed on EgoExoLearn (Huang et al., 2024) (120 hours of paired ego/exo videos). We instantiate each benchmark as a sequential task stream and consider three training regimes: Ego-only, Exo-only, and Ego+Exo co-training. Configurations are

summarized in Sec. 3.1 and Table 1. We report both within-view performance and zero-shot cross-view transfer.

**Baseline methods.** We implement a variety of representative CL methods including replay-based ER (Rolnick et al., 2019) and DER++ (Buzzega et al., 2020); regularization-based EWC (Kirkpatrick et al., 2017) and LwF (Li & Hoiem, 2017); and ensemble-based L2P (Wang et al., 2022d) and S-Prompt (Wang et al., 2022a; 2023b). Note that the replay-based methods ER and DER++ explicitly **reuse old training samples**, which are **typically unavailable** under restricted CL settings and are potentially unfair when compared to other replay-free methods. We adapt L2P and S-Prompt to CE⁴L by replacing prompts with adapters (Wang et al., 2023b) while preserving their original routing strategies (i.e., learnable task embeddings and $k$NN routing). We denote these variants as L2P+ and S-Prompt+. We also report joint training (upper bound) and sequential fine-tuning (lower bound). In all tables, we use light shading to group method families for readability. Scenario-specific details are provided in Appendix B.

**Evaluation.** After each task, we evaluate on all seen tasks and summarize by final average performance $\bar{A}_T$ and average forgetting $\bar{F}_T$ when applicable. We use test splits when available (otherwise the official validation split; Appendix B) and average over 3 seeds (± std). Best/second-best are highlighted per column, excluding joint training.

*Table 6.* Overall performance of representative CL methods on the **continual action anticipation**. We report the final average Top-5 recall (%,↑) for verb and noun prediction evaluated on egocentric and exocentric videos. -V, predict verb category. -N, predict noun category.

| Method | Train: Ego | | | | Train: Exo | | | | Train: Ego+Exo | | | |
|---|---|---|---|---|---|---|---|---|---|---|---|---|
| | Ego-V | Ego-N | Exo-V | Exo-N | Ego-V | Ego-N | Exo-V | Exo-N | Ego-V | Ego-N | Exo-V | Exo-N |
| Joint | $33.40_{\pm0.03}$ | $29.88_{\pm0.16}$ | $32.68_{\pm0.03}$ | $29.32_{\pm0.11}$ | $35.15_{\pm0.05}$ | $28.94_{\pm0.05}$ | $33.09_{\pm0.08}$ | $28.29_{\pm0.09}$ | $35.50_{\pm0.41}$ | $27.80_{\pm0.17}$ | $33.12_{\pm0.29}$ | $27.11_{\pm0.12}$ |
| Fine-tuning | $29.83_{\pm1.05}$ | $20.77_{\pm0.99}$ | $30.47_{\pm0.85}$ | $20.34_{\pm0.81}$ | $31.55_{\pm1.49}$ | $20.33_{\pm0.90}$ | $30.87_{\pm0.98}$ | $19.87_{\pm0.77}$ | $31.25_{\pm1.15}$ | $19.26_{\pm0.67}$ | $30.30_{\pm0.91}$ | $18.82_{\pm0.54}$ |
| ER | $31.15_{\pm0.49}$ | $26.47_{\pm0.47}$ | $\underline{30.84}_{\pm0.23}$ | $\mathbf{26.50}_{\pm0.21}$ | $\underline{34.39}_{\pm0.39}$ | $\mathbf{26.01}_{\pm0.36}$ | $\underline{32.77}_{\pm0.09}$ | $\underline{25.56}_{\pm0.21}$ | $\underline{34.49}_{\pm0.53}$ | $\mathbf{25.29}_{\pm0.24}$ | $\underline{32.70}_{\pm0.15}$ | $\underline{24.75}_{\pm0.23}$ |
| DER++ | $\underline{31.31}_{\pm0.36}$ | $\underline{26.91}_{\pm0.69}$ | $30.68_{\pm0.22}$ | $\underline{26.16}_{\pm0.56}$ | $34.02_{\pm0.33}$ | $25.54_{\pm0.78}$ | $32.44_{\pm0.05}$ | $25.20_{\pm0.60}$ | $33.70_{\pm0.37}$ | $24.55_{\pm0.53}$ | $32.37_{\pm0.26}$ | $24.13_{\pm0.35}$ |
| EWC | $29.72_{\pm1.12}$ | $20.63_{\pm1.00}$ | $30.16_{\pm0.37}$ | $20.19_{\pm0.84}$ | $31.69_{\pm1.43}$ | $20.41_{\pm0.89}$ | $30.63_{\pm1.17}$ | $19.95_{\pm0.75}$ | $31.29_{\pm1.04}$ | $19.96_{\pm0.48}$ | $30.64_{\pm0.95}$ | $19.44_{\pm0.25}$ |
| LwF | $29.84_{\pm0.77}$ | $22.24_{\pm0.77}$ | $29.75_{\pm0.75}$ | $21.79_{\pm0.74}$ | $30.47_{\pm0.71}$ | $21.17_{\pm0.55}$ | $30.27_{\pm0.30}$ | $20.67_{\pm0.35}$ | $30.57_{\pm0.21}$ | $21.38_{\pm1.46}$ | $31.19_{\pm1.32}$ | $20.58_{\pm1.16}$ |
| L2P+ | $28.76_{\pm1.14}$ | $16.24_{\pm1.18}$ | $27.88_{\pm1.20}$ | $15.50_{\pm1.34}$ | $30.03_{\pm2.24}$ | $16.57_{\pm1.14}$ | $28.06_{\pm1.82}$ | $15.59_{\pm1.17}$ | $28.51_{\pm1.89}$ | $16.19_{\pm0.80}$ | $26.12_{\pm0.93}$ | $15.21_{\pm0.62}$ |
| S-Prompt+ | $30.41_{\pm1.11}$ | $18.36_{\pm1.72}$ | $27.99_{\pm1.62}$ | $18.23_{\pm1.35}$ | $32.64_{\pm3.07}$ | $16.80_{\pm2.62}$ | $28.44_{\pm2.41}$ | $16.82_{\pm2.40}$ | $31.83_{\pm2.24}$ | $16.49_{\pm2.43}$ | $27.64_{\pm1.46}$ | $16.36_{\pm2.02}$ |
| VISTA (ours) | $\mathbf{32.37}_{\pm4.74}$ | $\mathbf{28.08}_{\pm2.15}$ | $\mathbf{34.83}_{\pm6.71}$ | $25.12_{\pm2.15}$ | $\mathbf{35.37}_{\pm3.69}$ | $\underline{25.69}_{\pm1.57}$ | $\mathbf{34.46}_{\pm3.63}$ | $\mathbf{26.40}_{\pm1.53}$ | $\mathbf{35.23}_{\pm1.05}$ | $\underline{25.23}_{\pm1.33}$ | $\mathbf{32.77}_{\pm3.37}$ | $\mathbf{26.01}_{\pm1.48}$ |

*Table 7.* **Ablation study of core design choices in VISTA on CE⁴L.** We evaluate component combinations of subspace router and task adapters under the Ego+Exo setting across all four CE⁴L scenarios. All numbers are reported as average performance after the last task.

| Subspace Router | Task Adapters | Skill RAAN (↑) | Segment (Ego+Exo) | | Association (Ego+Exo) | | Planning (Ego+Exo) | | Anticipation (Ego+Exo) | | | |
|---|---|---|---|---|---|---|---|---|---|---|---|---|
| | | | Ego (↑) | Exo (↑) | Ego→Exo (↑) | Exo→Ego (↑) | Ego (↓) | Exo (↓) | Ego-V (↑) | Ego-N (↑) | Exo-V (↑) | Exo-N (↑) |
| ✗ | ✗ | $72.94_{\pm2.15}$ | $35.51_{\pm1.82}$ | $20.32_{\pm1.75}$ | $41.13_{\pm1.99}$ | $35.57_{\pm0.17}$ | $85.53_{\pm0.56}$ | $82.68_{\pm1.21}$ | $31.25_{\pm1.15}$ | $19.26_{\pm0.67}$ | $30.30_{\pm0.91}$ | $18.82_{\pm0.54}$ |
| Random | ✓ | $69.41_{\pm1.27}$ | $38.03_{\pm1.82}$ | $26.02_{\pm3.80}$ | $41.00_{\pm0.36}$ | $36.90_{\pm0.92}$ | $84.06_{\pm0.06}$ | $82.99_{\pm0.98}$ | $28.66_{\pm1.76}$ | $16.01_{\pm3.10}$ | $28.15_{\pm1.93}$ | $16.22_{\pm2.96}$ |
| ✓ | ✓ | $80.38_{\pm0.32}$ | $45.35_{\pm1.34}$ | $31.41_{\pm0.87}$ | $44.74_{\pm1.18}$ | $38.29_{\pm1.16}$ | $82.81_{\pm0.21}$ | $80.74_{\pm0.92}$ | $35.23_{\pm1.05}$ | $25.23_{\pm1.33}$ | $32.77_{\pm3.37}$ | $26.01_{\pm1.48}$ |
| Oracle | ✓ | $80.42_{\pm0.51}$ | $47.17_{\pm3.16}$ | $33.23_{\pm2.11}$ | $45.07_{\pm0.50}$ | $41.87_{\pm0.87}$ | $82.35_{\pm0.27}$ | $81.04_{\pm1.35}$ | $35.35_{\pm4.18}$ | $22.57_{\pm5.06}$ | $35.45_{\pm3.75}$ | $23.70_{\pm4.83}$ |

## 5.2. Experimental Results

**Continual skill assessment.** Table 2 reports ranking accuracy and forgetting for RAAN and two reference-aware variants. Fine-tuning forgets substantially and falls short of joint training. Replay helps, with DER++ the strongest replay baseline, but still does not fully recover joint training. Regularization-based baselines are less effective: EWC largely tracks fine-tuning with high forgetting, while LwF is inconsistent across variants, sometimes suppressing forgetting but not translating into stronger final accuracy. In contrast, VISTA achieves joint-level accuracy on all variants while maintaining near-zero forgetting. This suggests that, for pairwise ranking, stabilizing updates via parameter penalties or distillation alone is insufficient. It benefits from parameter isolation with robust test-time task routing.

**Continual action segmentation.** Table 3 reveals a sharp separation between two aspects: for within-view, most CL methods improve clearly over fine-tuning, showing that interference is a major factor for dense parsing; for cross-view, zero-shot transfer remains low even for joint training, pointing to viewpoint-conditioned mismatch beyond CL. Ego+Exo co-training improves the cross-view evaluation substantially, suggesting that multi-view evidence is the primary lever for transferable segmentation problem, while current CL methods mainly control within-view retention and ignore such generalization ability.

**Continual cross-view association.** Table 4 shows that Ego+Exo co-training improves both retrieval directions remarkably, highlighting the important role of paired cross-view evidence for alignment. Under single-view CL, replay and regularization provide only modest, direction-dependent

gains over fine-tuning. The any-time curves (Figs. 6–8) show that S-Prompt+ and VISTA can improve over time with near-zero or negative forgetting, while curves for most other methods remain relatively flat, consistent with generally small forgetting observed for such contrastive objective (Cha et al., 2021; Ni et al., 2023; Tan et al., 2025), suggesting that task-wise statistics stabilize cross-view alignment while leveraging the evolving contrastive backbone. However, the remaining gap to joint training highlights the necessity of multi-view evidence for robust alignment, motivating future CL methods to better exploit such information.

**Continual action anticipation & planning.** Table 6 reports Top-5 recall for verb and noun prediction. In anticipation, ensemble-based baselines do not translate to consistent gains: L2P+ and S-Prompt+ are generally weaker than replay and VISTA, and their degradation is especially visible on noun prediction, suggesting that their routing signals do not reliably capture the task structure. Replay-based methods improve performance but only partially offsets the degradation, while regularization-based methods largely track fine-tuning. VISTA is consistently competitive across training regimes and label heads, improving both within-view and cross-view accuracy, which suggests that task-conditioned adapters can preserve complementary temporal cues across tasks without an explicit learned task classifier.

For planning (Table 5), regularization-based and replay-based baselines become competitive, and the gap to joint training is smaller. This suggests that, in our setup, planning is comparatively less dominated by catastrophic forgetting and more limited by the base representation of long-horizon procedural structure. VISTA remains consistently strong across views and training regimes, matching or improving

*Table 8.* **Generalization to future tasks during continual learning.** We report the upper-triangle average performance $\tilde{A}$ under the Ego+Exo setting, averaging over both views when applicable. Higher is better except planning ED@8 ($\downarrow$).

| Method | Skill | Seg. | Assoc. | Plan. | Ant.-V | Ant.-N |
|---|---|---|---|---|---|---|
| Fine-tuning | 47.69 | 18.95 | 30.09 | 88.55 | 22.48 | 12.50 |
| ER | 47.37 | 18.11 | 28.82 | 87.69 | 24.29 | 14.90 |
| DER++ | 47.51 | 18.84 | 30.70 | 87.93 | 26.10 | 14.85 |
| LwF | 46.43 | 20.02 | 31.09 | 87.52 | 24.94 | 14.96 |
| S-Prompt+ | 44.61 | 19.92 | 26.21 | 87.97 | 20.97 | 12.45 |
| VISTA (ours) | **48.04** | **24.53** | **31.80** | 87.18 | **26.89** | **15.04** |

*Table 9.* **Generalization to excluded tasks.** We evaluate tasks not used in the default CE4L Ego+Exo pipeline. E, Ego. X, Exo.

| Method | Segment | | Association | | Planning | |
|---|---|---|---|---|---|---|
| | Ego | Exo | E→X | X→E | Ego | Exo |
| Fine-tuning | 46.60 | 36.58 | 6.00 | 14.00 | 84.00 | 82.96 |
| ER | 42.54 | 38.28 | 6.00 | 17.00 | 84.11 | 79.00 |
| DER++ | 47.92 | 40.70 | 10.00 | 15.00 | 84.55 | 78.15 |
| LwF | 41.29 | 41.18 | 8.00 | 13.00 | **83.45** | 78.10 |
| S-Prompt+ | 47.54 | 38.13 | **17.00** | 17.00 | 84.01 | 80.14 |
| VISTA (ours) | **48.36** | **42.58** | **17.00** | **19.00** | 83.97 | **77.14** |

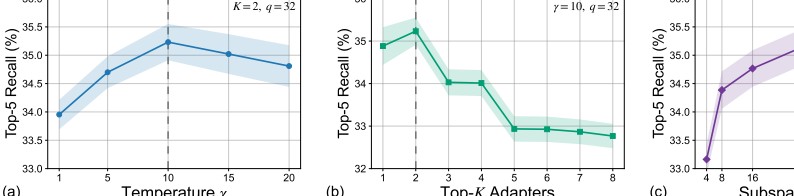

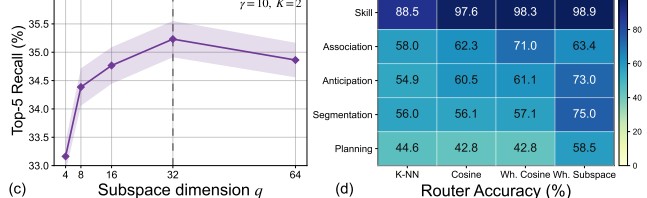

*Figure 3.* **Hyper-parameter sensitivity and routing strategy analysis.** (a) Different temperature parameter $\gamma$ for residual score computation, (b) top-$K$ choice of adapter mixture and (c) subspace dimension $q$. (d) Routing strategy comparison between $k$NN, cosine similarity, whitened cosine similarity and whitened subspace router. We report the final average top-5 recall ($\%, \uparrow$) of the continual anticipation trained on Ego+Exo data, evaluated on egocentric verb prediction (a-c), and task-router accuracy ($\%, \uparrow$) of VISTA (d).

upon other CL baselines while avoiding large replay buffers.

**Generalization beyond seen tasks.** The default CE4L protocol evaluates the union of seen tasks, while table 8 reports performance on future tasks before they are introduced into the continual stream, using the upper-triangle average of the performance matrix. Table 9 evaluates EgoExoLearn tasks excluded from the default CE4L stream. Across both settings, VISTA achieves the strongest overall performance in most cases, consistent with its robust subspace-based routing under inputs outside the seen-task evaluation set.

We report complementary metrics, validation results, additional forgetting analyses, backbone variants, and supplementary generality studies in Appendix B–D.

**Ablation and sensitivity study.** Table 7 isolates the roles of task adapters and routing. Removing both components reduces VISTA to sequential fine-tuning and yields the weakest performance across CE4L. Adding task adapters with a random router often degrades performance, indicating that naive mixture selection can induce systematic mis-routing and reintroduce interference despite parameter isolation. In contrast, the whitened-subspace router combined with task adapters consistently recovers strong performance across all scenarios. An oracle router (100% routing accuracy) provides an upper bound and is close to our router on most metrics. Overall, these results indicate that CE4L requires not only storing task-specific adaptation, but also performing accurate, view-robust task inference at test time.

Fig. 3 further examines hyper-parameter sensitivity and router design choices. Performance is stable around the default configuration used in all main experiments, with a clear optimum at intermediate routing temperature and a moderate top-$K$ mixture size, indicating that VISTA benefits from confident (but not overly peaky) routing, and that mixing too many adapters can dilute task-specific updates. Increasing the task subspace dimension improves performance up to a moderate rank and then saturates, suggesting that a compact whitened subspace captures most task-discriminative variation needed for routing. Fig. 3(d) compares routing strategies and shows that whitening is consistently beneficial, while the whitened-subspace residual router achieves the highest router accuracy across benchmarks, validating the second-order task statistics over instance-level nearest neighbors for robust task inference under viewpoint shift.

**Detailed analysis.** We analyze the router behavior from two complementary angles: task separability in the saved subspace geometry and empirical routing accuracy. Fig. 5 visualizes the residual scores for skill assessment and shows that test samples attain systematically lower residuals under their ground-truth task subspace than under other tasks, explaining the near-perfect routing observed in this scenario. To quantify task-level geometry, we compute distances between tasks using a principal-angle discrepancy between their augmented bases (see Appendix C for details). Fig. 4 shows that separability varies substantially across benchmarks: skill tasks form well-separated subspaces, while segmentation and association exhibit much smaller inter-

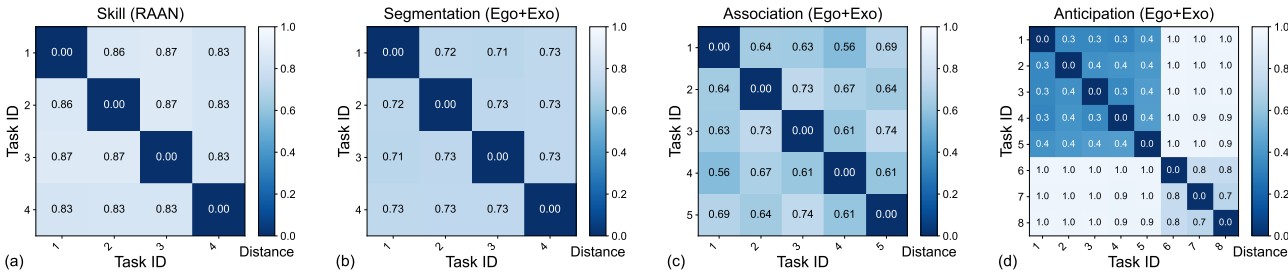

*Figure 4.* **Whitened-subspace distances across tasks.** (a-d) We visualize pairwise distances between task-specific whitened subspaces (spanned by $B_t$ with $q = 32$) used by the router (smaller distance means more similar) across 4 different scenarios.

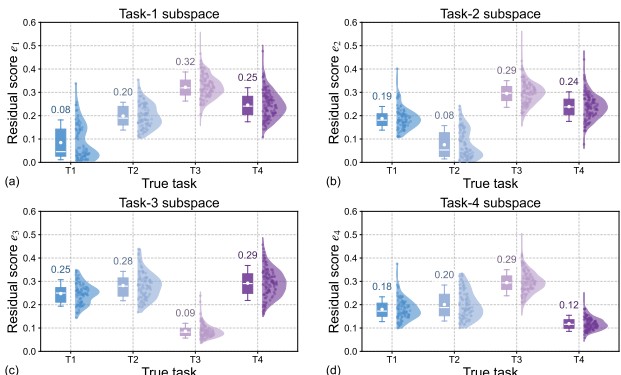

*Figure 5.* **Task-wise subspace residual score.** (a-d) We calculate the residual score of each test set feature in different task's subspace of continual skill assessment benchmark (RAAN).

*Table 10.* **Evaluation of efficiency on continual cross-view association benchmark.** We report the number of trainable and total parameters in millions, storage cost in million floats, and average computation time cost in millisecond per training batch.

| Method | Trainable (M) | Total (M) | Storage (M) | Time (ms) |
|---|---|---|---|---|
| Fine-tuning | 177.66 | 177.66 | 0.00 | 1023.5 |
| ER | 177.66 | 177.66 | 1696.67 | 947.3 |
| DER++ | 177.66 | 177.66 | 1698.22 | 1042.6 |
| EWC | 177.66 | 177.66 | 355.32 | 1516.8 |
| LwF | 177.66 | 177.66 | 177.66 | 1047.3 |
| L2P+ | 177.83 | 177.83 | 0.00 | 1425.4 |
| S-Prompt+ | 177.83 | 177.83 | 0.00 | 730.4 |
| VISTA | 177.83 | 177.83 | 0.01 | 793.2 |

task distances, indicating a more entangled task geometry. Anticipation further reveals a clustered structure where tasks in either kitchen or lab scenarios are mutually similar.

These geometric patterns align with routing difficulty in Fig. 3(d), but do not fully determine it. In the harder cases where task subspaces are closer, whitening and residual-energy scoring still recover informative task signals, improving router accuracy over cosine similarity and $k$-NN baselines. This suggests that routing benefits from task signatures that capture both mean shift and variability (first and second order), and that routing quality depends on both task separability and the metric used to compare. Appendix Tables 25–26 further compare GMM/Mahalanobis routers and adapter mixture choices.

Finally, Table 10 and Appendix Tables 27–30 compare and summarize methods' efficiency. Replay methods incur substantial memory storage due to buffering raw samples and additional targets for DER++. Regularization methods avoid large buffers but introduce extra state and computation for Fisher estimation or teacher snapshots. In contrast, VISTA adds only lightweight adapters and compact per-task statistics, keeping storage overhead negligible and per-batch compute close to fine-tuning. This makes VISTA a practical baseline for long streams and large backbones where replay buffers may be impractical or where resources are limited.

**Limitations and Scope.** CE$^4$L is instantiated on one ego-exo corpus and assumes tasks with fixed boundaries, focusing on embodied perception and semantic/procedural cross-view understanding rather than full 3D spatial reasoning or end-to-end decision making. VISTA stores one adapter and compact routing statistics per task, so long streams may benefit from adapter merging, compression, or hierarchical routing despite the small per-task overhead in our setting.

## 6. Conclusion

We introduced CE$^4$L, a benchmark suite for continual ego, exo, and ego-exo learning that evaluates continual learners under heterogeneous video objectives while explicitly separating within-view retention from cross-view generalization and multi-view co-training. Our empirical study shows that the effectiveness of classic continual paradigms varies substantially across problems, and that cross-view evaluation exposes viewpoint-conditioned mismatch that can persist even when within-view forgetting is reduced. To provide a strong and practical baseline in this regime, we proposed VISTA, a parameter-efficient continual learner with training-free adapter routing, which consistently improves performance and efficiency over baselines across CE$^4$L. As a general CL protocol, CE$^4$L can be instantiated on future ego-exo corpora with broader task coverage, additional viewpoints and sensors, and new supervision types, enabling CL research to scale with the rapidly evolving data landscape of embodied vision.

## Acknowledgment.

This work is supported by the NSFC Projects (Nos. 62406160, 62595773, 32530042), Beijing Natural Science Foundation (No. L247011), Beijing Major Science and Technology Project (No. Z251100008425003), and the STI2030-Major Projects (No. 2022ZD0204900).

## Impact Statement

This paper presents work whose goal is to advance the field of Machine Learning. There are many potential societal consequences of our work, none of which we feel must be specifically highlighted here.

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

# A. Background and Related Work

## A.1. Egocentric Video Learning

Egocentric vision research has transitioned from limited datasets (Pirsiavash & Ramanan, 2012; Bansal et al., 2022) to large-scale unscripted collections capturing complex daily activities (Damen et al., 2018; 2022; Liu et al., 2022). This domain recently integrated third-person perspectives to ground first-person experiences, a crucial development for understanding skilled interactions and procedural demonstrations (Kwon et al., 2021; Sigurdsson et al., 2018). Specialized benchmarks now target challenges such as asynchronous cross-view association (Sener et al., 2022; Xue & Grauman, 2023; Huang et al., 2024; Grauman et al., 2024; Huang et al., 2025), action quality assessment (Li et al., 2024; Ragusa et al., 2021; Zhou et al., 2024c; 2025a;b; 2026), and skill transfer to embodied agents via Vision-Language-Action models (Nair et al., 2022; Ma et al., 2022; Punamiya et al., 2025; Yang et al., 2025). Concurrently, representation learning has advanced toward unified backbones that directly fuse multiple modalities for robust zero-shot transfer (Lin et al., 2022; Pramanick et al., 2023; Zhao et al., 2023; Chen et al., 2022; Wang et al., 2022b; 2024b; 2025c;a). Related studies also examine static robustness and generalization under occlusions, corruptions, benchmark sensitivity, cross-view prediction, and audio-aided domain shifts (Grover et al., 2023; Schiappa et al., 2023; Thoker et al., 2022; Vyas et al., 2020; Zhang et al., 2022; Maharana et al., 2026). Despite these strides toward lifelong perception, current methods rely heavily on static offline training protocols (Plizzari et al., 2024; Mangalam et al., 2023). These standard approaches overlook the inherently non-stationary nature of visual streams where tasks and viewpoints evolve dynamically, a limitation this work directly addresses.

## A.2. Video Continual Learning

Video CL imposes unique demands due to the high-dimensional temporal dynamics absent in static image benchmarks (De Lange et al., 2021; Wang et al., 2024a; Zhou et al., 2024b). Early approaches largely extended image-based paradigms, including regularization and replay, by incorporating temporal constraints to mitigate catastrophic forgetting: Park et al. (2021) proposed time-channel importance maps to guide knowledge distillation, while vCLIMB (Villa et al., 2022) introduces temporal consistency regularization to optimize the selection of frame-level exemplars. Recently, the field has shifted towards parameter-efficient adaptation of large-scale pre-trained models (e.g., CLIP) to avoid the computational burden of full fine-tuning. Methods such as Space-time Prompting (Pei et al., 2023) and PIVOT (Villa et al., 2023) employ learnable prompts to capture task-specific temporal semantics while freezing the backbone, thereby enhancing transferability with minimal storage costs. Alternative strategies like STSP (Cheng et al., 2024) explore geometric constraints, utilizing orthogonal subspace projections to separate class representations without relying on privacy-invasive exemplars. Furthermore, the scope of video CL is expanding beyond simple action classification: recent benchmarks like ViLCo (Tang et al., 2024) and methods like Bisecle (Tan et al., 2025) investigate continuous adaptation in complex video-language understanding tasks or incorporate audio streams (Pian et al., 2023). However, these works predominantly focus on single-view streams or isolated recognition tasks, neglecting the complex coupled shifts in viewpoint and long-horizon procedural dependencies that our CE$^4$L seeks to bridge.

## A.3. Pretraining-based Continual Learning

Traditional CL studies are mostly conducted by training artificial neural networks from scratch over non-stationary data streams (Parisi et al., 2019; Wang et al., 2024a; Zhou et al., 2024b; Rolnick et al., 2019). With the rise of pretrained models (PTMs), the research paradigm of CL has shifted from unlocking the potential of small models to leveraging the powerful representational capabilities and generalization abilities of PTMs (Zhou et al., 2024a; Wang et al., 2022d). Theoretical studies of canonical CL have proved that it is possible to decompose the task-free CL problem into two orthogonal subproblems: task identity inference and within-task prediction (Kim et al., 2022). Yet the use of PTMs has changed the relation between these two subproblems, given rich and general prior information from pretraining data (Wang et al., 2023b). Recent works focus on adapting the frozen PTM backbone via parameter-efficient tuning (PET) techniques, in which lightweight trainable modules are inserted into the backbone's workflow (Wang et al., 2022d; Lester et al., 2021; Li & Liang, 2021; Rebuffi et al., 2017; Hu et al., 2021; Yan et al., 2025). Such PET components can be either task-shared (Wang et al., 2022c) or task-specific (Wang et al., 2022a; 2023b), and the latter often requires a routing strategy developed along the CL process. It is worth noting that a learnable router itself can suffer from catastrophic forgetting, and therefore a variety of work has proposed to address this issue using analytic or training-free algorithms (McDonnell et al., 2024; Zhuang et al., 2022). Following this direction, VISTA further investigates the efficacy of a non-parametric router for video data by describing the geometry of task-wise subspaces.

# B. Additional Benchmark Details and Results

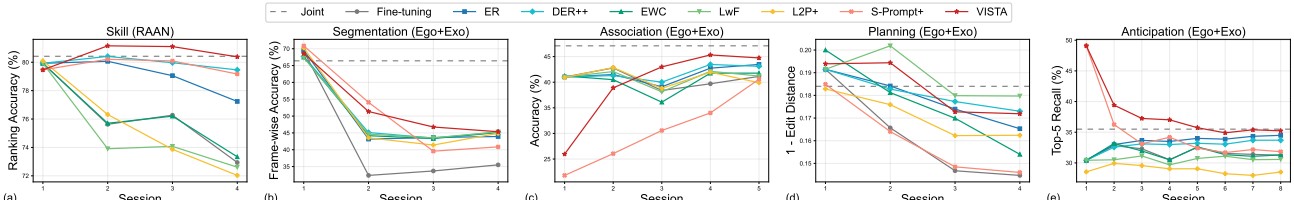

*Figure 6.* **CE⁴L performance curve (Ego + Exo).** (a-e) We visualize pairwise the performance curve throughout the complete CL progress across 5 different scenarios. Trained on egocentric + exocentric data, and evaluated on egocentric test data, except continual skill assessment using RAAN and only egocentric data for training.

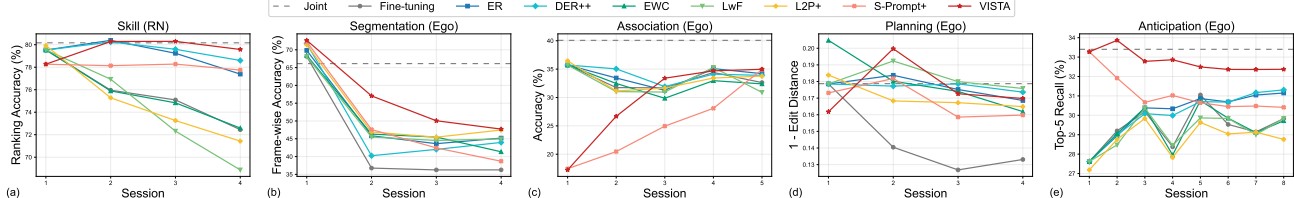

*Figure 7.* **CE⁴L performance curve (Ego).** (a-e) We visualize pairwise the performance curve throughout the complete CL progress across 5 different scenarios. Trained on egocentric data, and evaluated on egocentric test data, except skill assessment using relation network (RN) and both egocentric training data and exocentric reference data.

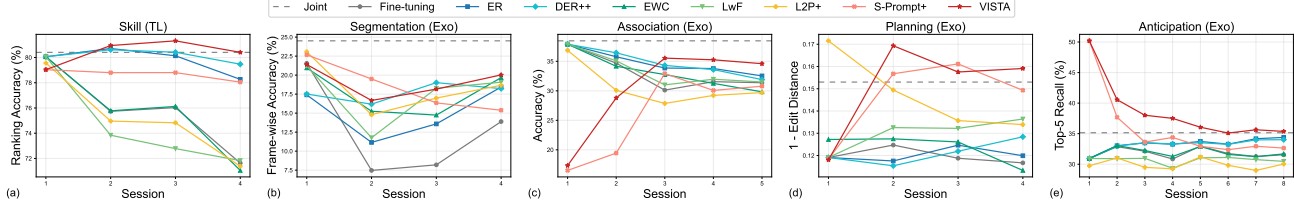

*Figure 8.* **CE⁴L performance curve (Exo).** (a-e) We visualize pairwise the performance curve throughout the complete CL progress across 5 different scenarios. Trained on exocentric data, and evaluated on egocentric test data, except continual skill assessment: triplet loss (TL) and both egocentric training data and exocentric reference data.

**Continual evaluation metrics.** We evaluate continual learners by recording a performance matrix $A \in \mathbb{R}^{T \times T}$, where $A_{t,j}$ denotes the performance on task $j$ after learning up to task $t$ ($1 \leq j \leq t \leq T$). The *final average performance* is defined as

$$\bar{A}_T = \frac{1}{T} \sum_{j=1}^{T} A_{T,j}. \tag{9}$$

The *average forgetting* summarizes how much performance on past tasks drops from its best historical value to the final value:

$$\bar{F}_T = \frac{1}{T-1} \sum_{j=1}^{T-1} \left( \max_{t \in \{j,...,T\}} A_{t,j} - A_{T,j} \right). \tag{10}$$

We use the benchmark-specific metric $A_{t,j}$ (e.g., ranking accuracy, frame-wise accuracy, Top-1 association accuracy, Top-5 recall, and ED@8) and report $\bar{A}_T$ and $\bar{F}_T$ when forgetting is meaningful for the task. For action planning where ED@8 is lower-is-better, we use the monotone transformation $1 - \text{ED@8}$ so that larger values always indicate better performance. We also report forward transfer as

$$\overline{\text{FWT}} = \frac{1}{T-1} \sum_{t=2}^{T} (A_{t-1,t} - A_{0,t}), \tag{11}$$

where $A_{0,t}$ denotes the pretrained model performance on task $t$ before continual training.

*Table 11.* **Detailed CE$^4$L benchmark statistics.** CE$^4$L is built on EgoExoLearn, which contains 8 high-level procedural tasks and 745 videos. Split format is train / val / test unless otherwise noted. Skill assessment uses train / val pairs only; association evaluation uses 800 validation and 2,000 test direction-balanced multiple-choice questions. All videos are standardized to 25 fps and 224×224 resolution.

| Scenario | Tasks | Task Unit | Ego Split | Exo Split / Input |
|---|---|---|---|---|
| Skill assessment | 4 | Action groups | 13,722 / 10,518 pairs | – / 10 segments per clip |
| Action segmentation | 4 | Procedural tasks | 173 / 28 / 55 videos | 176 / 21 / 26 videos; full video, 5× downsampled |
| Cross-view association | 5 | Procedural tasks | 22.7k clips (314 videos) | 5.4k clips (249 videos); 4 frames per clip |
| Action anticipation | 8 | Procedural tasks | 34.6k / 7.7k / 17.3k clips | 6.1k / 2.1k / 4.7k clips; 2 s context, 5 frames at 5 fps |
| Action planning | 4 | Procedural tasks | 2,171 / 395 / 734 clips | 1,847 / 205 / 270 clips; 2 s context, 5 frames at 5 fps |

*Table 12.* **Forward transfer on CE$^4$L.** We report $\overline{\text{FWT}}$ under Ego+Exo training. Higher is better for all metrics.

| Method | Skill | Segmentation | Association | Planning | Anticipation-V | Anticipation-N |
|---|---|---|---|---|---|---|
| Fine-tuning | **3.70** | 19.89 | 9.85 | 0.09 | 2.83 | 1.57 |
| ER | 2.93 | 18.53 | 0.33 | 0.81 | **4.07** | **3.01** |
| DER++ | 3.10 | 20.04 | 2.80 | 1.44 | 3.28 | 2.36 |
| LwF | 2.64 | 20.55 | 9.88 | 0.99 | 2.61 | 2.32 |
| S-Prompt+ | 1.18 | 22.98 | 6.64 | **2.14** | 3.06 | 2.31 |
| VISTA (ours) | 3.11 | **27.68** | **11.66** | 1.57 | 3.61 | 2.48 |

We show the any-time average performance curve of all CL methods across all CE$^4$L benchmarks in figures 6, 7 and 8. For the future-task evaluation used in Table 8, we compute $\tilde{A} = \frac{1}{T-1} \sum_{i=1}^{T-1} \left( \frac{1}{T-i} \sum_{j=i+1}^{T} A_{i,j} \right)$, the upper-triangle mean of the performance matrix before those tasks are learned. The excluded-task evaluation in Table 9 uses EgoExoLearn tasks not included in the default CE$^4$L stream. In the following sections, we provide detailed descriptions of the specific training settings for each benchmark, the baseline frameworks, the adaptation of all CL algorithms, and additional results.

**Shared continual baseline settings.** Unless otherwise noted, we use a single set of baseline hyper-parameters across all benchmarks. For ER (Rolnick et al., 2019) and DER++ (Buzzega et al., 2020), we set the buffer ratio to 0.2, meaning the replay memory capacity is 20% of the total number of training samples observed in the entire stream, and we set the replay batch ratio to 0.2, meaning 20% of each training mini-batch is replaced by replay samples while keeping the effective batch size unchanged. For DER++, we use a distillation weight of 0.5. For EWC (Kirkpatrick et al., 2017), we use a regularization weight of $10^{-2}$, an online Fisher decay of 1.0, and estimate the diagonal Fisher from 50 mini-batches at the end of each task. For LwF (Li & Hoiem, 2017), we use an MSE distillation weight of 0.5. For L2P+ (Wang et al., 2022d), we use a top-2 adapter selection and an adapter bottleneck dimension of 64. For S-Prompt+ (Wang et al., 2022a), we use hard top-1 routing with 32 KMeans centroids per task. For VISTA, we use a top-2 adapter mixture, an adapter bottleneck dimension of 64, router pooling with a single temporal segment, a subspace rank of 32, and a routing temperature of 10.0. Benchmark-specific training hyper-parameters and method-specific targets are given below.

### B.1. Continual Skill Assessment

**Training setup.** For the baseline architecture of continual skill assessment, we follow the framework of Huang et al. (2024), in which a margin ranking loss, a disparity loss and a rank-aware loss are utilized as the training objective for the ego-only baseline variant RAAN (Doughty et al., 2019); and for the ego+exo case given exocentric video reference, a triplet-loss based baseline variant TL is implemented by taking the reference video as anchor, and a relation-network baseline variant RN is introduced by respectively concatenating the two egocentric video features and exocentric feature (Sung et al., 2018). We train for 2000 epochs per task with Adam using a learning rate of $10^{-4}$ and batch size 128.

**Baseline method details.**

*Replay methods (ER and DER++).* We follow the shared replay protocol above. In skill assessment, DER++ caches the pair of aggregated ranking scores $(s_1, s_2)$ produced at the time a sample is written to memory (averaged over RAAN branches and time steps), and uses an MSE distillation loss with weight 0.5 on replayed samples.

*Regularization methods (EWC and LwF).* We follow the shared settings above. For EWC, we estimate a diagonal Fisher

from 50 mini-batches at each task end and use a regularization weight of $10^{-2}$ with online decay 1.0. For LwF, we distill toward a teacher snapshot via an MSE loss on ranking scores with weight 0.5.

*Ensemble-based baselines (L2P+ and S-Prompt+).* L2P+ maintains a fixed pool of adapters and learns a key vector per adapter. For each input pair, it builds a query representation by temporally pooling the clip features into a single segment and selects the top-2 adapters by cosine match between the query and the keys. The selected adapters are applied and averaged, and an auxiliary similarity loss encourages stable key query matching. S-Prompt+ uses hard top-1 routing based on task-wise KMeans centroids: after each task, it clusters task features into 32 centers and assigns a test input to the single nearest task by the minimum average distance to that task's centers, then applies only the selected task adapter. A key implementation difference is whether the shared predictor is frozen: for S-Prompt+ and VISTA, adapters are explicitly task-indexed and past adapters are not revisited once the stream moves on, so we freeze the shared learner (i.e., the prediction head) after the first task to keep earlier adapters calibrated; in contrast, L2P+ draws from a shared adapter pool whose members can continue to receive updates across tasks, so it does not require freezing the shared learner.

*Table 13.* **TAS results on the validation split (frame-wise accuracy).** We report the final average frame-wise accuracy $\bar{A}_T$ (%, ↑) for egocentric and exocentric evaluation under each training regime.

| Method | Train: Ego | | Train: Exo | | Train: Ego+Exo | |
|---|---|---|---|---|---|---|
| | Eval: Ego | Eval: Exo | Eval: Ego | Eval: Exo | Eval: Ego | Eval: Exo |
| Joint | $70.51_{\pm0.73}$ | $21.40_{\pm0.99}$ | $26.99_{\pm1.67}$ | $40.06_{\pm1.38}$ | $71.25_{\pm0.35}$ | $40.60_{\pm1.16}$ |
| Fine-tuning | $45.41_{\pm2.15}$ | $8.21_{\pm3.15}$ | $22.85_{\pm1.53}$ | $28.98_{\pm2.31}$ | $46.14_{\pm2.19}$ | $28.13_{\pm1.41}$ |
| ER | $45.11_{\pm2.62}$ | $10.94_{\pm5.76}$ | $\underline{22.98}_{\pm2.40}$ | $29.68_{\pm1.66}$ | $44.23_{\pm5.03}$ | $29.89_{\pm1.29}$ |
| DER++ | $45.48_{\pm3.43}$ | $7.46_{\pm2.93}$ | $18.65_{\pm5.59}$ | $\underline{31.63}_{\pm2.48}$ | $43.95_{\pm2.55}$ | $30.59_{\pm3.29}$ |
| EWC | $41.23_{\pm9.09}$ | $6.86_{\pm0.31}$ | $21.31_{\pm5.84}$ | $29.98_{\pm3.12}$ | $45.88_{\pm0.85}$ | $31.65_{\pm2.04}$ |
| LwF | $46.04_{\pm3.57}$ | $\underline{11.38}_{\pm1.21}$ | $22.19_{\pm2.76}$ | $31.13_{\pm0.99}$ | $44.85_{\pm2.49}$ | $\mathbf{32.76}_{\pm2.29}$ |
| L2P+ | $\underline{47.77}_{\pm1.64}$ | $7.89_{\pm0.81}$ | $18.36_{\pm4.52}$ | $29.24_{\pm1.65}$ | $\underline{46.18}_{\pm2.38}$ | $30.93_{\pm2.83}$ |
| S-Prompt+ | $40.26_{\pm1.06}$ | $6.52_{\pm0.75}$ | $15.42_{\pm2.45}$ | $26.35_{\pm1.74}$ | $42.62_{\pm1.58}$ | $23.17_{\pm1.40}$ |
| VISTA (ours) | $\mathbf{50.57}_{\pm2.08}$ | $\mathbf{12.80}_{\pm3.19}$ | $\mathbf{24.98}_{\pm3.46}$ | $\mathbf{33.73}_{\pm0.86}$ | $\mathbf{49.56}_{\pm0.81}$ | $\underline{32.13}_{\pm2.66}$ |

*Table 14.* **TAS forgetting on the test split (frame-wise accuracy).** We report the final average forgetting $\bar{F}_T$ (%, ↓) of frame-wise accuracy for both view evaluation under each training regime.

| Method | Train: Ego | | Train: Exo | | Train: Ego+Exo | |
|---|---|---|---|---|---|---|
| | Eval: Ego | Eval: Exo | Eval: Ego | Eval: Exo | Eval: Ego | Eval: Exo |
| Joint | $--$ | $--$ | $--$ | $--$ | $--$ | $--$ |
| Fine-tuning | $33.12_{\pm1.94}$ | $12.98_{\pm9.56}$ | $\mathbf{-0.55}_{\pm1.19}$ | $14.21_{\pm2.20}$ | $31.53_{\pm5.69}$ | $15.61_{\pm3.01}$ |
| ER | $34.12_{\pm4.56}$ | $9.29_{\pm9.56}$ | $0.04_{\pm3.00}$ | $11.33_{\pm2.47}$ | $33.94_{\pm6.57}$ | $14.31_{\pm2.84}$ |
| DER++ | $32.27_{\pm2.77}$ | $8.88_{\pm2.62}$ | $5.08_{\pm5.18}$ | $12.01_{\pm3.45}$ | $32.90_{\pm4.55}$ | $14.39_{\pm5.78}$ |
| EWC | $36.89_{\pm8.48}$ | $11.45_{\pm4.42}$ | $1.31_{\pm6.01}$ | $8.96_{\pm3.67}$ | $29.88_{\pm3.16}$ | $13.74_{\pm2.19}$ |
| LwF | $33.83_{\pm5.44}$ | $8.53_{\pm0.75}$ | $1.46_{\pm4.32}$ | $12.04_{\pm2.63}$ | $32.38_{\pm3.63}$ | $13.77_{\pm2.74}$ |
| L2P+ | $30.11_{\pm2.20}$ | $13.41_{\pm2.20}$ | $3.47_{\pm5.83}$ | $11.37_{\pm1.04}$ | $29.49_{\pm2.47}$ | $11.51_{\pm1.88}$ |
| S-Prompt+ | $\underline{11.13}_{\pm0.14}$ | $\underline{7.24}_{\pm8.00}$ | $3.77_{\pm0.44}$ | $\mathbf{3.14}_{\pm0.65}$ | $\underline{10.15}_{\pm3.94}$ | $\underline{5.55}_{\pm3.50}$ |
| VISTA (ours) | $\mathbf{3.56}_{\pm0.22}$ | $\mathbf{6.42}_{\pm4.66}$ | $\underline{-0.09}_{\pm1.10}$ | $\underline{3.70}_{\pm0.40}$ | $\mathbf{2.81}_{\pm0.74}$ | $\mathbf{1.68}_{\pm0.62}$ |

*Table 15.* **TAS results on the test split (edit score).** We report the final average edit score $\bar{A}_T$ (↑) for egocentric and exocentric evaluation under each training regime.

| Method | Train: Ego | | Train: Exo | | Train: Ego+Exo | |
|---|---|---|---|---|---|---|
| | Eval: Ego | Eval: Exo | Eval: Ego | Eval: Exo | Eval: Ego | Eval: Exo |
| Joint | $46.34_{\pm0.58}$ | $19.57_{\pm2.83}$ | $33.93_{\pm3.42}$ | $34.22_{\pm0.72}$ | $43.41_{\pm0.42}$ | $34.77_{\pm2.71}$ |
| Fine-tuning | $42.04_{\pm0.91}$ | $24.65_{\pm2.14}$ | $27.68_{\pm4.51}$ | $28.53_{\pm3.04}$ | $40.38_{\pm1.59}$ | $28.80_{\pm3.08}$ |
| ER | $42.69_{\pm1.00}$ | $22.95_{\pm2.12}$ | $\mathbf{28.82}_{\pm1.64}$ | $32.95_{\pm3.35}$ | $38.66_{\pm6.29}$ | $30.88_{\pm3.70}$ |
| DER++ | $42.51_{\pm2.25}$ | $25.19_{\pm0.86}$ | $\underline{27.79}_{\pm2.77}$ | $35.29_{\pm2.30}$ | $38.32_{\pm2.43}$ | $29.02_{\pm1.13}$ |
| EWC | $40.35_{\pm5.21}$ | $25.07_{\pm1.11}$ | $27.49_{\pm2.18}$ | $33.44_{\pm0.51}$ | $\underline{40.85}_{\pm2.23}$ | $30.26_{\pm2.24}$ |
| LwF | $\underline{43.10}_{\pm2.11}$ | $\underline{26.77}_{\pm0.50}$ | $26.31_{\pm1.42}$ | $34.63_{\pm2.57}$ | $\mathbf{41.19}_{\pm1.10}$ | $29.37_{\pm0.50}$ |
| L2P+ | $42.85_{\pm0.64}$ | $24.77_{\pm4.77}$ | $11.03_{\pm0.62}$ | $36.12_{\pm2.32}$ | $39.59_{\pm4.69}$ | $33.64_{\pm1.91}$ |
| S-Prompt+ | $36.66_{\pm1.97}$ | $15.46_{\pm6.86}$ | $14.49_{\pm1.26}$ | $\underline{36.76}_{\pm1.43}$ | $31.79_{\pm1.62}$ | $\underline{34.58}_{\pm3.82}$ |
| VISTA (ours) | $\mathbf{44.88}_{\pm0.90}$ | $\mathbf{27.90}_{\pm5.91}$ | $27.74_{\pm3.41}$ | $\mathbf{37.94}_{\pm1.93}$ | $39.83_{\pm6.23}$ | $\mathbf{36.26}_{\pm0.37}$ |

*Table 16.* **TAS results on the test split (segmental F1@Avg).** We report the final average segmental F1@Avg $\bar{A}_T$ (↑) for egocentric and exocentric evaluation under each training regime.

| Method | Train: Ego | | Train: Exo | | Train: Ego+Exo | |
|---|---|---|---|---|---|---|
| | Eval: Ego | Eval: Exo | Eval: Ego | Eval: Exo | Eval: Ego | Eval: Exo |
| Joint | $41.60_{\pm0.15}$ | $7.01_{\pm0.80}$ | $8.75_{\pm0.76}$ | $18.50_{\pm0.56}$ | $38.91_{\pm0.88}$ | $18.95_{\pm0.92}$ |
| Fine-tuning | $30.63_{\pm1.08}$ | $6.46_{\pm1.05}$ | $6.85_{\pm1.48}$ | $14.15_{\pm0.86}$ | $27.19_{\pm1.54}$ | $14.40_{\pm1.37}$ |
| ER | $31.40_{\pm0.66}$ | $5.35_{\pm0.54}$ | $\mathbf{7.67}_{\pm0.91}$ | $14.76_{\pm1.29}$ | $27.87_{\pm3.50}$ | $14.65_{\pm1.55}$ |
| DER++ | $\underline{32.02}_{\pm0.64}$ | $6.44_{\pm0.74}$ | $7.09_{\pm1.00}$ | $15.25_{\pm1.59}$ | $27.44_{\pm1.80}$ | $14.65_{\pm1.55}$ |
| EWC | $30.12_{\pm4.01}$ | $5.75_{\pm1.39}$ | $7.19_{\pm1.14}$ | $\underline{15.49}_{\pm1.11}$ | $28.49_{\pm1.96}$ | $\underline{15.35}_{\pm1.94}$ |
| LwF | $31.46_{\pm1.10}$ | $\mathbf{7.43}_{\pm0.31}$ | $\underline{7.42}_{\pm0.57}$ | $15.38_{\pm1.02}$ | $\mathbf{29.40}_{\pm1.64}$ | $13.74_{\pm1.87}$ |
| L2P+ | $30.41_{\pm0.09}$ | $7.07_{\pm1.56}$ | $4.45_{\pm0.09}$ | $13.19_{\pm1.01}$ | $27.97_{\pm3.16}$ | $14.21_{\pm1.39}$ |
| S-Prompt+ | $22.49_{\pm0.62}$ | $3.76_{\pm1.53}$ | $5.16_{\pm0.25}$ | $12.63_{\pm0.22}$ | $19.17_{\pm1.06}$ | $12.61_{\pm0.94}$ |
| VISTA (ours) | $\mathbf{33.26}_{\pm0.78}$ | $\underline{7.23}_{\pm1.27}$ | $6.19_{\pm0.47}$ | $\mathbf{15.68}_{\pm0.57}$ | $\underline{29.27}_{\pm4.14}$ | $\mathbf{15.39}_{\pm0.68}$ |

*Table 17.* **TAS results on the validation split (edit score).** We report the final average edit score $\bar{A}_T$ (↑) for egocentric and exocentric evaluation under each training regime.

| Method | Train: Ego | | Train: Exo | | Train: Ego+Exo | |
|---|---|---|---|---|---|---|
| | Eval: Ego | Eval: Exo | Eval: Ego | Eval: Exo | Eval: Ego | Eval: Exo |
| Joint | $47.91_{\pm0.52}$ | $22.44_{\pm1.98}$ | $34.58_{\pm3.82}$ | $37.32_{\pm0.63}$ | $44.72_{\pm1.27}$ | $37.64_{\pm1.89}$ |
| Fine-tuning | $43.17_{\pm2.31}$ | $25.41_{\pm2.59}$ | $27.47_{\pm3.35}$ | $31.11_{\pm2.44}$ | $39.95_{\pm1.28}$ | $29.56_{\pm1.17}$ |
| ER | $\underline{44.74}_{\pm1.80}$ | $24.15_{\pm2.09}$ | $\mathbf{28.97}_{\pm1.69}$ | $35.42_{\pm2.81}$ | $38.26_{\pm5.47}$ | $30.12_{\pm2.24}$ |
| DER++ | $43.72_{\pm1.59}$ | $25.03_{\pm1.52}$ | $26.31_{\pm1.81}$ | $\underline{36.15}_{\pm0.83}$ | $37.94_{\pm3.00}$ | $30.04_{\pm2.86}$ |
| EWC | $40.04_{\pm5.68}$ | $24.75_{\pm1.29}$ | $26.66_{\pm2.04}$ | $32.69_{\pm0.28}$ | $40.18_{\pm1.74}$ | $\underline{34.25}_{\pm1.33}$ |
| LwF | $44.62_{\pm0.86}$ | $\underline{27.24}_{\pm1.10}$ | $25.20_{\pm1.66}$ | $35.66_{\pm3.52}$ | $\mathbf{42.52}_{\pm0.18}$ | $32.70_{\pm1.69}$ |
| L2P+ | $44.04_{\pm0.78}$ | $26.49_{\pm2.43}$ | $10.85_{\pm0.69}$ | $31.81_{\pm0.92}$ | $40.25_{\pm4.30}$ | $33.73_{\pm2.42}$ |
| S-Prompt+ | $36.66_{\pm0.35}$ | $16.44_{\pm3.66}$ | $14.38_{\pm1.48}$ | $32.65_{\pm0.96}$ | $31.35_{\pm1.72}$ | $32.74_{\pm3.10}$ |
| VISTA (ours) | $\mathbf{45.35}_{\pm1.38}$ | $\mathbf{28.87}_{\pm3.52}$ | $27.90_{\pm3.27}$ | $\mathbf{37.42}_{\pm0.99}$ | $\underline{42.51}_{\pm6.58}$ | $\mathbf{37.11}_{\pm1.40}$ |

*Table 18.* **TAS results on the validation split (segmental F1@Avg).** We report the final average segmental F1@Avg $\bar{A}_T$ (↑) for egocentric and exocentric evaluation under each training regime.

| Method | Train: Ego | | Train: Exo | | Train: Ego+Exo | |
|---|---|---|---|---|---|---|
| | Eval: Ego | Eval: Exo | Eval: Ego | Eval: Exo | Eval: Ego | Eval: Exo |
| Joint | $44.94_{\pm0.85}$ | $8.61_{\pm0.37}$ | $8.56_{\pm0.52}$ | $21.07_{\pm0.39}$ | $41.56_{\pm0.14}$ | $22.34_{\pm0.92}$ |
| Fine-tuning | $29.82_{\pm1.71}$ | $5.64_{\pm0.88}$ | $5.94_{\pm0.86}$ | $16.46_{\pm2.45}$ | $25.26_{\pm0.32}$ | $15.62_{\pm1.90}$ |
| ER | $\underline{32.05}_{\pm1.44}$ | $5.37_{\pm0.69}$ | $\mathbf{6.42}_{\pm0.84}$ | $16.14_{\pm0.84}$ | $27.09_{\pm2.15}$ | $14.55_{\pm0.60}$ |
| DER++ | $31.86_{\pm0.78}$ | $5.07_{\pm1.27}$ | $5.73_{\pm0.89}$ | $\underline{17.28}_{\pm0.29}$ | $27.48_{\pm1.97}$ | $15.30_{\pm1.40}$ |
| EWC | $29.28_{\pm3.40}$ | $4.56_{\pm0.22}$ | $\underline{6.41}_{\pm1.03}$ | $15.62_{\pm1.46}$ | $27.89_{\pm0.57}$ | $\mathbf{16.55}_{\pm1.69}$ |
| LwF | $30.62_{\pm0.42}$ | $\mathbf{6.78}_{\pm1.09}$ | $6.07_{\pm0.12}$ | $15.85_{\pm0.88}$ | $\underline{29.02}_{\pm0.13}$ | $15.95_{\pm0.68}$ |
| L2P+ | $30.86_{\pm1.29}$ | $4.91_{\pm0.21}$ | $4.01_{\pm0.09}$ | $11.91_{\pm0.79}$ | $27.29_{\pm2.00}$ | $13.84_{\pm2.19}$ |
| S-Prompt+ | $23.26_{\pm1.18}$ | $3.57_{\pm1.12}$ | $4.59_{\pm0.23}$ | $10.98_{\pm1.05}$ | $19.46_{\pm0.85}$ | $12.06_{\pm0.54}$ |
| VISTA (ours) | $\mathbf{34.89}_{\pm1.44}$ | $\underline{5.87}_{\pm1.25}$ | $5.81_{\pm0.55}$ | $\mathbf{18.69}_{\pm0.20}$ | $\mathbf{30.72}_{\pm3.60}$ | $\underline{16.52}_{\pm1.62}$ |

## B.2. Continual Action Segmentation

**Training setup.** For the baseline architecture of continual action segmentation, we follow the framework of Huang et al. (2024) and Chen et al. (2020), in which a per-stage training loss, consisting of a classification loss and a smoothing loss, is used as the objective for MS-TCN (Farha & Gall, 2019) model. For experimental efficiency, the frame sequence of both training and inference input video $v$ is down-sampled by a factor of 5. We train for 150 epochs per task with Adam using a learning rate of $5 \times 10^{-4}$ and batch size 1 video.

**Baseline method details.** We follow the shared baseline settings above. A replay item stores the feature tensor, frame-wise labels, and validity mask; DER++ caches masked last-stage source logits and distills them on replayed samples. EWC and LwF are applied to the same masked prediction tensor. Adapters act on per-frame features before MS-TCN. And we set temporal pooling parameter $M = 2$ for VISTA because the input length of segmentation is longer.

**Additional results.** We additionally report validation performance, test performance on complementary metrics, and test forgetting for continual temporal action segmentation in Tabs. 13–18.

## B.3. Continual Cross-View Association

**Training setup.** For the baseline architecture of continual cross-view association benchmark of CE⁴L, we follow the framework of Huang et al. (2024), whose dual-encoder backbone consisting of a TimeSformer-B (Bertasius et al., 2021) video encoder and a CLIP text encoder (Radford et al., 2021) is initialized with EgoVLP (Lin et al., 2022) weights pretrained on Ego4D (Grauman et al., 2022) dataset. A contrastive loss between video embedding and annotation text embedding is used to fine-tune the backbone model during training, while upon inference, only the video encoder is used to extract features for query and candidate videos. The prediction is given by choosing one from 20 candidates, which has the highest cosine similarity with the query. We train for 5 epochs per task with AdamW using a learning rate of $10^{-5}$ and batch size 32.

**Baseline method details.** We follow the shared baseline settings above. For DER++, we cache per-sample embedding targets consisting of the video embedding and the text embedding, and distill them on replayed samples. LwF distills both embeddings toward the teacher snapshot with weight 0.5. Adapter-based routing methods act on the video embedding. And we set the temporal pooling hyperparameter $M = 2$ for VISTA because the input length of association videos is longer.

**Additional results.** Additional validation results and test forgetting for continual association are reported in Tabs. 19 and 20.

*Table 19.* **Association results on the validation split.** We report the final average Top-1 accuracy $\bar{A}_T$ (%,↑) for Ego→Exo and Exo→Ego retrieval under each training regime.

| Method | Train: Ego | | Train: Exo | | Train: Ego+Exo | |
|---|---|---|---|---|---|---|
| | Ego→Exo | Exo→Ego | Ego→Exo | Exo→Ego | Ego→Exo | Exo→Ego |
| Joint | $35.17_{\pm0.72}$ | $40.83_{\pm0.62}$ | $39.00_{\pm1.08}$ | $40.08_{\pm0.72}$ | $43.08_{\pm1.23}$ | $45.83_{\pm1.45}$ |
| Fine-tuning | $25.25_{\pm1.77}$ | $26.50_{\pm2.65}$ | $33.25_{\pm1.87}$ | $35.92_{\pm1.16}$ | $36.25_{\pm0.74}$ | $37.00_{\pm2.76}$ |
| ER | $28.67_{\pm1.56}$ | $30.75_{\pm0.41}$ | $31.50_{\pm2.13}$ | $35.83_{\pm0.42}$ | $\underline{39.25}_{\pm0.94}$ | $39.50_{\pm2.67}$ |
| DER++ | $\underline{29.08}_{\pm1.01}$ | $30.25_{\pm0.54}$ | $31.58_{\pm2.86}$ | $35.08_{\pm0.12}$ | $38.17_{\pm0.62}$ | $\mathbf{40.83}_{\pm2.57}$ |
| EWC | $27.17_{\pm2.58}$ | $26.58_{\pm1.71}$ | $29.25_{\pm2.56}$ | $31.42_{\pm0.12}$ | $36.75_{\pm1.81}$ | $37.75_{\pm2.41}$ |
| LwF | $25.67_{\pm2.62}$ | $26.08_{\pm3.60}$ | $\underline{33.75}_{\pm1.06}$ | $37.08_{\pm1.55}$ | $35.83_{\pm0.47}$ | $36.67_{\pm2.79}$ |
| L2P+ | $\underline{29.08}_{\pm0.72}$ | $26.42_{\pm1.78}$ | $32.75_{\pm1.43}$ | $\mathbf{37.42}_{\pm1.85}$ | $36.92_{\pm2.47}$ | $37.08_{\pm1.01}$ |
| S-Prompt+ | $29.00_{\pm1.95}$ | $\mathbf{32.33}_{\pm0.77}$ | $31.08_{\pm2.04}$ | $34.83_{\pm0.47}$ | $35.58_{\pm0.51}$ | $39.08_{\pm0.31}$ |
| VISTA (ours) | $\mathbf{31.04}_{\pm2.95}$ | $\mathbf{32.70}_{\pm0.25}$ | $\mathbf{35.52}_{\pm5.24}$ | $\underline{37.33}_{\pm4.71}$ | $\mathbf{40.70}_{\pm5.24}$ | $\underline{40.00}_{\pm0.00}$ |

*Table 20.* **Association forgetting on the test split.** We report the final average forgetting $\bar{F}_T$ (%,↓) for Ego→Exo and Exo→Ego retrieval under each training regime.

| Method | Train: Ego | | Train: Exo | | Train: Ego+Exo | |
|---|---|---|---|---|---|---|
| | Ego→Exo | Exo→Ego | Ego→Exo | Exo→Ego | Ego→Exo | Exo→Ego |
| Joint | $--$ | $--$ | $--$ | $--$ | $--$ | $--$ |
| Fine-tuning | $3.09_{\pm2.46}$ | $8.25_{\pm3.59}$ | $6.54_{\pm1.53}$ | $2.50_{\pm3.02}$ | $2.27_{\pm1.96}$ | $2.69_{\pm1.55}$ |
| ER | $2.37_{\pm0.94}$ | $6.29_{\pm2.82}$ | $5.37_{\pm1.48}$ | $2.10_{\pm2.94}$ | $0.19_{\pm0.92}$ | $0.66_{\pm3.19}$ |
| DER++ | $2.98_{\pm0.80}$ | $6.96_{\pm1.78}$ | $6.04_{\pm1.28}$ | $3.04_{\pm2.52}$ | $1.23_{\pm2.81}$ | $2.85_{\pm1.63}$ |
| EWC | $3.32_{\pm1.58}$ | $9.92_{\pm1.84}$ | $8.07_{\pm3.05}$ | $4.40_{\pm3.37}$ | $1.94_{\pm2.15}$ | $1.49_{\pm1.99}$ |
| LwF | $5.63_{\pm5.18}$ | $8.32_{\pm4.32}$ | $6.44_{\pm2.25}$ | $1.79_{\pm3.77}$ | $2.50_{\pm2.34}$ | $3.00_{\pm1.26}$ |
| L2P+ | $2.68_{\pm0.65}$ | $8.58_{\pm1.61}$ | $7.16_{\pm2.17}$ | $5.80_{\pm2.44}$ | $4.42_{\pm1.03}$ | $2.96_{\pm1.92}$ |
| S-Prompt+ | $\underline{-6.84}_{\pm3.27}$ | $\mathbf{-7.42}_{\pm2.17}$ | $\underline{2.81}_{\pm1.42}$ | $\underline{-3.75}_{\pm1.43}$ | $\underline{-6.54}_{\pm4.73}$ | $\mathbf{-3.70}_{\pm3.68}$ |
| VISTA (ours) | $\mathbf{-7.20}_{\pm0.51}$ | $\underline{-6.39}_{\pm2.78}$ | $\mathbf{2.41}_{\pm1.78}$ | $\mathbf{-4.05}_{\pm0.82}$ | $\mathbf{-6.58}_{\pm0.78}$ | $\underline{-3.40}_{\pm0.70}$ |

## B.4. Continual Action Anticipation & Planning

**Training setup.** For the baseline architecture of continual action anticipation and planning, we follow the framework of Damen et al. (2022); Grauman et al. (2022) and Huang et al. (2024), the model is structured as a trainable temporal relation module TA3N (Chen et al., 2019) over a pretrained CLIP encoder (Radford et al., 2021) which is frozen. For both anticipation and planning, the training objectives are cross-entropy based classification losses, but one is calculated over one action step and the other is evaluated over an action sequence. The evaluation metrics for anticipation and planning are Top-5 recall and edit distance at 8 steps (ED@8). We train for 40 epochs per task with SGD using a learning rate of 0.003. The batch size is 128 for anticipation and 16 for planning.

**Baseline method details.** We follow the shared baseline settings above. DER++ caches the output logits as distillation targets and distills them on replayed samples. LwF distills output logits toward the teacher snapshot.

**Additional results.**

*Table 21.* **Anticipation forgetting.** We report the final average forgetting $\bar{F}_T$ $(\%, \downarrow)$ of Top-5 recall for verb and noun prediction evaluated on egocentric and exocentric videos under each training regime.

| Method | Train: Ego | | | | Train: Exo | | | | Train: Ego+Exo | | | |
|---|---|---|---|---|---|---|---|---|---|---|---|---|
| | Ego-V | Ego-N | Exo-V | Exo-N | Ego-V | Ego-N | Exo-V | Exo-N | Ego-V | Ego-N | Exo-V | Exo-N |
| Joint | $--$ | $--$ | $--$ | $--$ | $--$ | $--$ | $--$ | $--$ | $--$ | $--$ | $--$ | $--$ |
| Fine-tuning | $2.82_{\pm0.80}$ | $10.41_{\pm0.29}$ | $2.49_{\pm0.31}$ | $10.55_{\pm0.55}$ | $5.79_{\pm0.76}$ | $8.36_{\pm0.28}$ | $4.49_{\pm1.55}$ | $8.41_{\pm0.74}$ | $5.76_{\pm0.50}$ | $8.61_{\pm0.26}$ | $4.62_{\pm1.35}$ | $8.47_{\pm0.51}$ |
| ER | $0.59_{\pm0.23}$ | $2.30_{\pm0.60}$ | $1.17_{\pm0.64}$ | $2.25_{\pm0.70}$ | $1.49_{\pm0.63}$ | $1.63_{\pm0.62}$ | $1.33_{\pm0.68}$ | $1.78_{\pm0.45}$ | $1.59_{\pm0.40}$ | $1.33_{\pm0.60}$ | $\mathbf{0.74}_{\pm0.50}$ | $1.12_{\pm0.45}$ |
| DER++ | $\underline{0.19}_{\pm0.35}$ | $\underline{1.43}_{\pm0.55}$ | $0.46_{\pm0.48}$ | $\underline{1.57}_{\pm0.76}$ | $\underline{0.88}_{\pm0.29}$ | $\underline{1.04}_{\pm0.45}$ | $0.98_{\pm0.36}$ | $\underline{1.29}_{\pm0.57}$ | $\underline{0.79}_{\pm0.40}$ | $\underline{0.53}_{\pm0.45}$ | $\underline{0.76}_{\pm0.38}$ | $\underline{0.62}_{\pm0.52}$ |
| EWC | $3.02_{\pm1.06}$ | $10.34_{\pm0.58}$ | $2.82_{\pm0.99}$ | $10.64_{\pm0.83}$ | $5.67_{\pm0.94}$ | $8.47_{\pm0.32}$ | $4.82_{\pm1.31}$ | $8.43_{\pm0.63}$ | $5.64_{\pm0.54}$ | $7.85_{\pm0.29}$ | $3.88_{\pm1.65}$ | $7.79_{\pm0.24}$ |
| LwF | $1.41_{\pm0.89}$ | $7.63_{\pm0.23}$ | $2.36_{\pm0.54}$ | $7.83_{\pm0.45}$ | $3.61_{\pm0.28}$ | $7.08_{\pm0.86}$ | $3.91_{\pm0.61}$ | $7.04_{\pm0.90}$ | $3.37_{\pm0.73}$ | $5.68_{\pm0.91}$ | $3.60_{\pm0.98}$ | $5.76_{\pm0.80}$ |
| L2P+ | $1.99_{\pm1.03}$ | $3.30_{\pm2.14}$ | $2.99_{\pm0.71}$ | $3.73_{\pm1.93}$ | $2.01_{\pm0.90}$ | $3.04_{\pm0.92}$ | $2.22_{\pm0.65}$ | $3.61_{\pm1.05}$ | $2.10_{\pm0.85}$ | $4.44_{\pm1.00}$ | $2.83_{\pm1.25}$ | $4.87_{\pm1.05}$ |
| S-Prompt+ | $0.35_{\pm0.22}$ | $2.10_{\pm2.17}$ | $\underline{0.43}_{\pm0.54}$ | $2.41_{\pm2.59}$ | $1.92_{\pm1.13}$ | $1.87_{\pm1.83}$ | $2.97_{\pm2.78}$ | $2.01_{\pm2.04}$ | $1.34_{\pm0.65}$ | $1.41_{\pm1.20}$ | $1.14_{\pm1.02}$ | $1.56_{\pm1.42}$ |
| VISTA (ours) | $\mathbf{0.07}_{\pm0.05}$ | $\mathbf{0.76}_{\pm0.53}$ | $\mathbf{0.37}_{\pm0.22}$ | $\mathbf{0.80}_{\pm0.59}$ | $\mathbf{0.44}_{\pm0.17}$ | $\mathbf{0.53}_{\pm0.45}$ | $\mathbf{0.85}_{\pm0.96}$ | $\mathbf{0.62}_{\pm0.51}$ | $\mathbf{0.33}_{\pm0.20}$ | $\mathbf{0.46}_{\pm0.54}$ | $1.24_{\pm0.93}$ | $\mathbf{0.49}_{\pm0.57}$ |

*Table 22.* **Planning forgetting.** We report the final average forgetting $\bar{F}_T$ $(\%, \downarrow)$ computed on $1 - \text{ED}@8$.

| Method | Train: Ego | | Train: Exo | | Train: Ego+Exo | |
|---|---|---|---|---|---|---|
| | Eval: Ego | Eval: Exo | Eval: Ego | Eval: Exo | Eval: Ego | Eval: Exo |
| Joint | $--$ | $--$ | $--$ | $--$ | $--$ | $--$ |
| Fine-tuning | $2.26_{\pm0.09}$ | $3.71_{\pm0.90}$ | $0.52_{\pm0.08}$ | $4.23_{\pm1.58}$ | $3.33_{\pm0.74}$ | $6.30_{\pm1.42}$ |
| ER | $2.63_{\pm0.89}$ | $2.58_{\pm0.46}$ | $1.30_{\pm0.75}$ | $3.40_{\pm0.95}$ | $3.60_{\pm0.69}$ | $2.78_{\pm0.53}$ |
| DER++ | $1.46_{\pm0.43}$ | $1.77_{\pm0.59}$ | $\underline{-0.04}_{\pm0.69}$ | $2.16_{\pm0.71}$ | $1.59_{\pm0.22}$ | $1.39_{\pm1.47}$ |
| EWC | $4.74_{\pm0.61}$ | $0.97_{\pm0.95}$ | $1.79_{\pm0.31}$ | $3.39_{\pm1.65}$ | $4.96_{\pm0.38}$ | $4.17_{\pm1.23}$ |
| LwF | $1.71_{\pm0.73}$ | $0.86_{\pm0.59}$ | $0.21_{\pm0.89}$ | $4.12_{\pm0.45}$ | $1.96_{\pm1.00}$ | $2.76_{\pm0.64}$ |
| L2P+ | $2.61_{\pm1.40}$ | $1.85_{\pm0.63}$ | $2.73_{\pm1.94}$ | $3.22_{\pm2.02}$ | $3.27_{\pm2.12}$ | $5.70_{\pm4.54}$ |
| S-Prompt+ | $\mathbf{-0.04}_{\pm0.95}$ | $\underline{0.37}_{\pm0.19}$ | $1.60_{\pm0.37}$ | $\underline{0.45}_{\pm1.60}$ | $\underline{1.25}_{\pm0.48}$ | $\underline{1.02}_{\pm0.91}$ |
| VISTA (ours) | $\underline{0.53}_{\pm1.15}$ | $\mathbf{0.36}_{\pm1.03}$ | $\mathbf{-0.26}_{\pm1.16}$ | $\mathbf{0.15}_{\pm1.15}$ | $\mathbf{0.87}_{\pm0.49}$ | $\mathbf{0.66}_{\pm1.14}$ |

## B.5. Backbone Variants

We further evaluate the same CE$^4$L protocol with EgoVLPv2 (Pramanick et al., 2023) and VideoMAEv2 (Wang et al., 2023a) backbones. Tables 23 and 24 show that the relative behavior of CL methods and VISTA's advantage are largely preserved across backbone choices, suggesting that the findings are not artifacts of a single encoder.

*Table 23.* **Additional results with EgoVLPv2 backbone.** We report the final average performance under the Ego+Exo setting. Higher is better except planning ED@8 ($\downarrow$).

| Method | Skill | Assoc. | | Planning | | Anticipation | | | |
|---|---|---|---|---|---|---|---|---|---|
| | RAAN | E→X | X→E | Ego | Exo | Ego-V | Ego-N | Exo-V | Exo-N |
| Fine-tuning | 65.08 | 26.20 | 20.30 | 82.91 | 84.71 | 16.47 | 20.81 | 16.62 | 18.61 |
| ER | 76.92 | 29.30 | 23.40 | 81.43 | 82.21 | 21.72 | 26.90 | 21.83 | 23.81 |
| DER++ | 77.08 | 32.20 | 28.70 | **79.50** | 79.97 | 22.46 | 29.95 | 22.63 | 25.29 |
| EWC | 70.20 | 31.20 | 29.00 | 82.54 | 82.26 | 17.62 | 21.65 | 17.85 | 19.23 |
| LwF | 69.05 | 32.40 | 30.10 | 80.78 | 81.66 | 18.71 | 26.30 | 18.96 | 22.63 |
| L2P+ | 76.66 | 31.00 | 29.10 | 81.70 | 81.89 | 19.22 | 23.21 | 19.34 | 20.22 |
| S-Prompt+ | 77.36 | 32.60 | 28.40 | 81.81 | 81.15 | 19.86 | 22.94 | 19.77 | 20.52 |
| VISTA (ours) | **78.18** | **33.60** | **30.70** | 79.72 | **79.42** | **23.72** | 28.12 | **23.71** | **25.94** |

*Table 24.* **Additional results with VideoMAEv2 backbone.** We report the final average performance under the Ego+Exo setting.

| Method | Skill | Planning | | Anticipation | | | |
|---|---|---|---|---|---|---|---|
| | RAAN | Ego | Exo | Ego-V | Ego-N | Exo-V | Exo-N |
| Fine-tuning | 71.15 | 81.59 | 82.04 | 20.38 | 26.24 | 20.35 | 22.69 |
| ER | 77.66 | 78.55 | 79.64 | 26.34 | 31.38 | 27.20 | 26.81 |
| DER++ | 78.46 | **77.72** | 78.64 | 26.78 | 31.26 | 27.67 | 26.82 |
| EWC | 76.02 | 80.79 | 80.50 | 21.26 | 28.18 | 22.29 | 24.07 |
| LwF | 77.96 | 78.54 | 81.18 | 23.05 | 31.76 | 23.02 | **28.33** |
| L2P+ | 79.88 | 78.90 | 79.72 | 22.49 | 28.59 | 23.72 | 24.30 |
| S-Prompt+ | 80.01 | 79.61 | 80.77 | 23.39 | 28.37 | 23.31 | 25.00 |
| VISTA (ours) | **81.67** | 78.36 | **78.31** | **27.87** | **32.12** | **28.84** | 27.83 |

## C. Additional VISTA Analysis

### C.1. Subspace Visualization

We quantify task-level geometry by comparing augmented whitened-subspace bases through principal-angle discrepancy. Given two orthonormal bases $B_i \in \mathbb{R}^{d \times r_i}$ and $B_j \in \mathbb{R}^{d \times r_j}$, we compute the singular values $\{\sigma_\ell\}$ of $B_i^\top B_j$, which satisfy $\sigma_\ell = \cos\theta_\ell$ where $\{\theta_\ell\}$ are the principal angles between the two subspaces. We then define the discrepancy

$$D_{ij} = \sqrt{\frac{1}{m}\sum_{\ell=1}^{m}\sin^2\theta_\ell}, \quad m = \min(r_i, r_j), \tag{12}$$

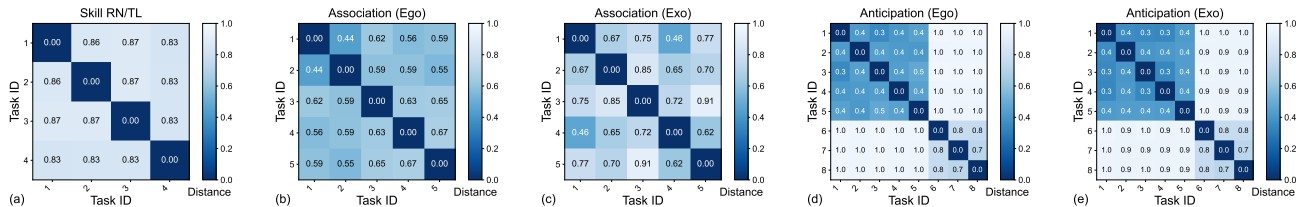

*Figure 9.* **Whitened-subspace distances across tasks.** (a-e) We visualize pairwise distances between task-specific subspaces (spanned by $B_t$ with $q = 32$) used by the router (smaller distance means more similar) across 5 different scenarios.

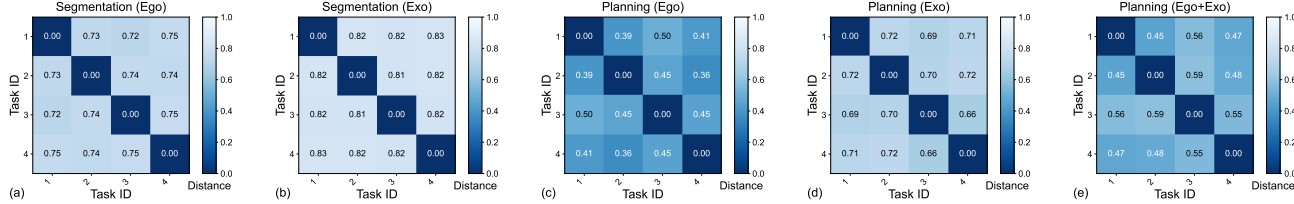

*Figure 10.* **Whitened-subspace distances across tasks.** (a-e) We visualize pairwise distances between task-specific subspaces (spanned by $B_t$ with $q = 32$) used by the router (smaller distance means more similar) across 5 different scenarios.

so that smaller $D_{ij}$ indicates more similar subspaces.

## C.2. Routing and Adapter Mixture Ablations

Table 25 compares the whitened-subspace router with first-order and density-based alternatives. First-order methods are strong on some short-clip cases but degrade when longer inputs and view-induced variance dominate; Gaussian Mixture Model (GMM) and Mahalanobis modeling help in selected scenarios but remain less consistent overall. Table 26 compares hard and soft adapter selection. Top-1 routing is sufficient for well-separated skill subspaces, while residual-weighted top-2 mixing is more reliable on harder benchmarks.

*Table 25.* **Router comparison on CE$^4$L.** We report task-router accuracy ($\%, \uparrow$) using the same task-specific adapters and only changing the routing rule. "Whitened" denotes applying the same variance normalization used by VISTA.

| Router | Skill | Segmentation | Association | Planning | Anticipation |
|---|---|---|---|---|---|
| $k$NN | 88.50 | 55.95 | 57.96 | 44.56 | 54.94 |
| Cosine | 97.58 | 56.14 | 62.25 | 42.78 | 60.55 |
| Whitened Cosine | 98.31 | 57.14 | **70.96** | 42.78 | 61.15 |
| GMM | 88.06 | 64.29 | 59.25 | 44.81 | 54.23 |
| Mahalanobis | 88.78 | 65.00 | 62.88 | 50.63 | 53.58 |
| Whitened Subspace (ours) | **98.86** | **75.00** | 63.42 | **58.48** | **73.04** |

*Table 26.* **Adapter mixture ablation across CE$^4$L.** We compare oracle routing, hard top-1 routing, unweighted top-2 averaging, and the residual-weighted top-2 mixture used by VISTA. Higher is better except planning ED@8 ($\downarrow$).

| Mixture Rule | Skill | Segmentation | | Association | | Planning | | Anticipation | | | |
|---|---|---|---|---|---|---|---|---|---|---|---|
| | RAAN | Ego | Exo | Ego→Exo | Exo→Ego | Ego | Exo | Ego-V | Ego-N | Exo-V | Exo-N |
| Oracle | **80.42** | **47.17** | **33.23** | **45.07** | **49.87** | 82.35 | 81.04 | **35.35** | 22.57 | **35.45** | 23.70 |
| Top-1 | 79.62 | 42.70 | 27.74 | 41.50 | 38.10 | 83.85 | 82.72 | 34.68 | 20.81 | 32.37 | 20.80 |
| Top-2 Avg. | 76.05 | 42.15 | 30.91 | 40.50 | 37.55 | 83.96 | 81.14 | 32.82 | 15.10 | 31.13 | 15.30 |
| Top-2 Mix (ours) | 80.38 | 45.35 | 31.41 | 44.74 | 38.29 | 82.81 | 80.74 | 35.23 | 25.23 | 32.77 | **26.01** |

## C.3. Computational Efficiency

All of our experiments are implemented using PyTorch on `Ubuntu 22.04 LTS`. The hardware environment consists of an `Intel Xeon Gold 6530` CPU and 8 `NVIDIA GeForce RTX 5090` GPUs. Notably, our proposed framework is efficient enough that each run requires only a single GPU for both training and inference. In the largest current VISTA

*Table 27.* **Evaluation of computational efficiency on continual skill assessment benchmark.** We report the number of trainable and total parameters in millions, storage cost in million floats, and average computation time cost in millisecond per training batch.

| Method | Trainable (M) | Total (M) | Storage (M) | Time (ms) |
|---|---|---|---|---|
| Fine-tuning | 1.84 | 1.84 | 0.00 | 32.7 |
| ER | 1.84 | 1.84 | 28.10 | 47.7 |
| DER++ | 1.84 | 1.84 | 28.10 | 49.6 |
| EWC | 1.84 | 1.84 | 3.68 | 126.8 |
| LwF | 1.84 | 1.84 | 1.84 | 71.9 |
| L2P+ | 2.38 | 2.38 | 0.00 | 67.6 |
| S-Prompt+ | 2.38 | 2.38 | 0.00 | 31.2 |
| VISTA | 2.38 | 2.38 | 0.04 | 34.7 |

*Table 28.* **Evaluation of efficiency on continual temporal action segmentation benchmark.** We report the number of trainable and total parameters in millions, storage cost in million floats, and average computation time cost in millisecond per training batch.

| Method | Trainable (M) | Total (M) | Storage (M) | Time (ms) |
|---|---|---|---|---|
| Fine-tuning | 0.74 | 0.74 | 0.00 | 395.0 |
| ER | 0.74 | 0.74 | 213.81 | 392.0 |
| DER++ | 0.74 | 0.74 | 219.50 | 364.4 |
| EWC | 0.74 | 0.74 | 1.48 | 1166.4 |
| LwF | 0.74 | 0.74 | 0.74 | 535.8 |
| L2P+ | 1.28 | 1.28 | 0.00 | 731.9 |
| S-Prompt+ | 1.28 | 1.28 | 0.00 | 359.7 |
| VISTA | 1.28 | 1.28 | 0.04 | 357.2 |

*Table 29.* Evaluation of computational efficiency on the **continual action planning benchmark.** We report the number of trainable and total parameters in millions, storage cost in million floats, and average computation time cost in millisecond per training batch.

| Method | Trainable (M) | Total (M) | Storage (M) | Time (ms) |
|---|---|---|---|---|
| Fine-tuning | 3.39 | 3.39 | 0.00 | 16.0 |
| ER | 3.39 | 3.39 | 6.97 | 91.5 |
| DER++ | 3.39 | 3.39 | 7.02 | 148.5 |
| EWC | 3.39 | 3.39 | 6.78 | 54.9 |
| LwF | 3.39 | 3.39 | 3.39 | 38.6 |
| L2P+ | 3.79 | 3.79 | 0.00 | 58.4 |
| S-Prompt+ | 3.79 | 3.79 | 0.00 | 25.0 |
| VISTA | 3.79 | 3.79 | 0.03 | 28.5 |

*Table 30.* **Evaluation of computational efficiency on continual action anticipation benchmark.** We report the number of trainable and total parameters in millions, storage cost in million floats, and average computation time cost in millisecond per training batch.

| Method | Trainable (M) | Total (M) | Storage (M) | Time (ms) |
|---|---|---|---|---|
| Fine-tuning | 3.24 | 3.24 | 0.00 | 19.3 |
| ER | 3.24 | 3.24 | 148.73 | 123.6 |
| DER++ | 3.24 | 3.24 | 149.46 | 169.0 |
| EWC | 3.24 | 3.24 | 6.47 | 83.4 |
| LwF | 3.24 | 3.24 | 3.24 | 29.8 |
| L2P+ | 4.05 | 4.05 | 0.00 | 37.7 |
| S-Prompt+ | 4.04 | 4.04 | 0.00 | 26.2 |
| VISTA | 4.04 | 4.04 | 0.03 | 19.9 |

setting, skill assessment uses $C = d = 1024$, $r = 64$, and $q = 32$, corresponding to about 0.17M scalars per task ($\approx 0.68$ MB in fp32). This is far smaller than replay storage such as ER on skill assessment and anticipation, which stores 56.20M and 148.73M scalars respectively (Tables 27 and 30).

# D. Additional Generality Analysis

Beyond CE$^4$L, we include two supplementary evaluations to investigate the applicability of VISTA. VISTA-Prompt replaces adapters with prompts on general CL image benchmarks under online blurry-boundary settings (Moon et al., 2023; Kang et al., 2025; Sun et al., 2026; Yan et al., 2026); results are shown in Table 31. We also report continual VLA results on LIBERO-10 (Liu et al., 2023; Römer et al., 2026) in Table 32, connecting the same design to embodied decision-making tasks. These experiments are not part of the main CE$^4$L benchmark, but they indicate that the routing principle is applicable beyond continual multi-view video perception.

*Table 31.* **VISTA-Prompt on general continual learning benchmarks.** We replace adapters with prompts and evaluate under online blurry-boundary settings. $A_{\mathrm{AUC}}$ denotes anytime accuracy AUC and $A_{\mathrm{last}}$ denotes final average accuracy. The backbone is ViT-B/16 pretrained on ImageNet-21K.

| Method | CIFAR-100 | | ImageNet-R | | CUB-200 | |
|---|---|---|---|---|---|---|
| | $A_{\mathrm{AUC}}$ | $A_{\mathrm{last}}$ | $A_{\mathrm{AUC}}$ | $A_{\mathrm{last}}$ | $A_{\mathrm{AUC}}$ | $A_{\mathrm{last}}$ |
| L2P | 76.23 | 79.11 | 44.40 | 42.03 | 64.30 | 61.42 |
| DualPrompt | 76.04 | 76.62 | 46.13 | 40.80 | 65.03 | 62.43 |
| CODA-Prompt | 79.13 | 80.91 | 51.87 | 48.09 | 66.01 | 62.90 |
| MISA | 80.35 | 80.75 | 51.52 | 45.08 | 65.40 | 60.20 |
| HiDe-Prompt | 77.10 | 81.77 | 53.77 | 49.87 | **67.05** | **67.12** |
| VISTA-Prompt (ours) | **81.10** | **83.52** | **54.50** | **52.47** | 65.03 | 64.72 |

*Table 32.* **Continual LIBERO-10 performance.** We report success-rate AUC, forward transfer (FWT), and negative backward transfer (NBT). The model uses DINOv2, CLIP text features, and DiT flow matching, pretrained on LIBERO-90. We follow the standard pipeline of previous work (Römer et al., 2026).

| Method | AUC (↑) | FWT (↑) | NBT (↓) |
|---|---|---|---|
| SeqFFT | 22.37 | 76.13 | 74.70 |
| SeqLoRA | 21.37 | 73.10 | 71.64 |
| PackNet | 4.84 | 37.20 | 41.34 |
| ER | 60.54 | **76.60** | 22.74 |
| CLARE | 74.07 | 74.27 | 0.18 |
| VISTA (ours) | **74.92** | 74.17 | **-0.39** |

## D.1. Pseudo Code

---

**Algorithm 1** VISTA: Video Incremental Subspace-routed Task Adapters

---

1: **Inputs**: task stream $\{\mathcal{D}_t\}_{t=1}^T$, backbone $f_\theta$, shared head $g_\psi$, subspace rank $q$, $\epsilon = 10^{-6}$, $\gamma$=10.0, top-$K$=2.

2:

3: **(1) Continual training and router fitting**

4: Initialize seen task set $\mathcal{S} \leftarrow \emptyset$

5: **for** $t = 1$ to $T$ **do**

6:     Train the current task adapter parameters $\phi_t$ on $\mathcal{D}_t$ (freeze $f_\theta$; freeze $g_\psi$ after task 1).

7:     $\mathcal{S} \leftarrow \mathcal{S} \cup \{t\}$

8:     Collect routing vectors $\{r_n\}_{n=1}^{N_t} \leftarrow \{r(f_\theta(x)) : x \in \mathcal{D}_t\}$.

9:     Compute mean and diagonal variance: $\mu_t \leftarrow \frac{1}{N_t}\sum_{n=1}^{N_t} r_n, \;\; \mathrm{var}_t \leftarrow \frac{1}{N_t}\sum_{n=1}^{N_t} r_n \odot r_n - \mu_t \odot \mu_t$.

10:     Whitening weights: $w_t \leftarrow \sqrt{\mathrm{var}_t + \epsilon}$.

11:     Centered whitened residuals: $z_n \leftarrow (r_n - \mu_t) \odot w_t$.

12:     Covariance: $\Sigma_t \leftarrow \frac{1}{N_t-1}\sum_{n=1}^{N_t} z_n z_n^\top$.

13:     Top-$q$ eigenvectors: $U_t \leftarrow \mathrm{eig}_q(\Sigma_t) \in \mathbb{R}^{d \times q}$.

14:     Whitened mean direction: $m_t \leftarrow \frac{\mu_t \odot w_t}{\|\mu_t \odot w_t\|_2}$.

15:     Augmented basis: $B_t \leftarrow \mathrm{QR}([m_t \; U_t]) \in \mathbb{R}^{d \times (q+1)}$.

16: **end for**

17:

18: **(2) Task-agnostic inference**

19: **for** test input $v$ **do**

20:     $x \leftarrow f_\theta(v); \quad r \leftarrow r(x)$.

21:     **for** each $t \in \mathcal{S}$ **do**

22:       $\tilde{r}_t \leftarrow r \odot w_t$.

23:       $e_t(r) \leftarrow 1 - \frac{\|B_t^\top \tilde{r}_t\|_2^2}{\|\tilde{r}_t\|_2^2 + \epsilon}$.

24:     **end for**

25:     $p_t(r) \leftarrow \mathrm{softmax}(-\gamma\, e(r))$.

26:     Select top-$K$=2 tasks $\{t_\ell\}_{\ell=1}^2$ with largest $p_t(r)$ and normalize $\{\pi_\ell\}$.

27:     Adapter mixture: $\tilde{x} \leftarrow \sum_{\ell=1}^2 \pi_\ell\, a_{\phi_{t_\ell}}(x)$.

28:     Prediction: $\hat{y} \leftarrow g_\psi(\tilde{x})$.

29: **end for**

---

