# OpenReview forum: "CE$^4$L: Continual Ego, Exo, and Ego-Exo Learning"
_ICML.cc/2026/Conference — ICML 2026 regular_

### Official Review · Reviewer_RMdJ · 2026-03-05

**Soundness:** 4
**Presentation:** 3
**Significance:** 3
**Originality:** 3
**Overall Recommendation:** 5
**Confidence:** 4

**Summary:**

This paper introduces a benchmark called Continual Ego, Exo, and Ego-Exo Learning ($CE^4L$) for continual learning in multi-view video understanding. This falls into embodied perception because embodied agents (e.g., robots) often receive both egocentric and exocentric video inputs. This benchmark contains four sequential tasks: continual pairwise skill assessment, temporal action segmentation, cross-view association, and action anticipation & planning. The metrics separately measure within-view retention, cross-view generalization. Additionally, this paper proposes Video Incremental Subspace-routed Task Adapters (VISTA), which is a parameter-efficient continual learning approach. It uses lightweight adapters to store task knowledge and training-free routing for continual multi-view video learning. Experiments compare many continual learning strategies including elastic weight consolidation (EWC) and results show that the proposed method VISTA achieves good to best performance in many metrics and it is a strong baseline for this benchmark.

**Compliance With Llm Reviewing Policy:**

Affirmed.

**Final Justification:**

The rebuttal fully addressed my concerns, hence increasing the rating.

**Key Questions For Authors:**

1. Many continual learning evaluation includes forward transfer metric (how learning task $T_t$ helps learning task $T_{t+1--n}$), why this is omitted in this benchmark?
2. Can these four tasks actually be meaningful for existing embodied decision-making tasks? For example, will a method that achieves high performance on this benchmark also achieve high performance on some existing embodied tasks?”

**Limitations:**

1. Extend the work to actual robotics and embodied agent tasks (e.g., navigation, manipulation, long-horizon task planning).

**Strengths And Weaknesses:**

Strength:
1. **Strong motivation and novel benchmark.** Traditional continual learning assumes exocentric images but embodied agents use egocentric views and sometimes use multiview visual streams. Therefore, $CE^4L$ has strong motivation for combining multiview video understanding and continual learning. Additionally, existing CL evaluation abstract away strong viewpoint bias and heterogeneous semantic objectives, and this work brings an evaluation protocol that explicitly models these challenges by organizing multi-view video tasks into a continual learning stream. Models must adapt to shifting viewpoints, different task objectives, and temporally structure. Thus, the evaluation protocol aims to more faithfully reflect real-world embodied learning scenarios, enabling the assessment of both knowledge retention across tasks and cross-view generalization under the framework of continual learning.
2. **Diverse tasks.** Connecting to the last point where existing CL doesn't have heterogeneous semantic objectives. This work has 4 diverse tasks in a sequential manner: continual pairwise skill assessment, temporal action segmentation, cross-view association, and action anticipation & planning. Multiple semantic objectives make the continual learning to be more challenging than a single task (e.g., classification).
3. **Solid experiments and interesting insights.** Authors bring many set of experiments as well as ablation study to show the importance of key designs in the proposed method VISTA, as well as a very detailed comparison of current methodologies and their weaknesses. There are some interesting insights too, for example, the cross-view evaluation exposes viewpoint-conditioned mismatch can persist when within-view forgetting is reduced. The comprehensive benchmark and the strong baseline can be very meaningful for the field of embodied continual learning.
4. **Good writing and clear figures.**

Weaknesses
1. The proposed method VISTA is relatively incremental because task-specific adaptors are commonly used in continual learning. The paper should state clearer about what's the key novelty here.

---

> ### Author Rebuttal · Authors · 2026-03-31
>
> We sincerely thank you for your constructive comments. We appreciate your recognition of our motivation, task diversity, solid experiments, and clear writing. Below we provide point-to-point responses to **Weaknesses (W)** and **Questions (Q)**, and will add corresponding revisions in the final version.
>
> ### **W1: VISTA's novelty relative to task-specific adapter methods.**
>
> VISTA's contribution is the training-free whitened-subspace residual router that sits on top of task-specific adapters. Existing adapter-based CL methods rely on learned routing: L2P trains a shared key pool, S-Prompt uses K-NN over trained key vectors or HiDe-Prompt costly retrains the task identifier at the end of every tasks with stored feature distributions. Most of such CL methods introduce learnable components for task inference that can themselves forget or drift. VISTA instead accumulates per-task second-order statistics (mean, diagonal variance, principal components) without any trainable routing parameters. Whitening normalizes viewpoint-dependent variance, and the subspace residual scores task affinity by how well a sample's variation pattern aligns with each task's principal directions. This provides a richer signal than mean-based matching and is better aligned with multi-view settings. The soft top-$K$ mixture further allows graceful interpolation near task boundaries rather than hard selection. We will revise the method introduction to foreground this distinction.
>
> ### **Q1: Forward transfer in continual evaluation.**
>
> Forward transfer was not omitted by design. The main paper focused on final average performance and forgetting as the primary axes, and the anytime curves implicitly reflect forward dynamics. We now compute FWT explicitly. Let $A_{i,j}$ denote the performance on task $j$ after learning up to task $i$, and $A_{0,j}$ the initial model's performance on task $j$ before any CL training. FWT is $\overline{\text{FWT}} = \frac{1}{T-1}\sum_{t=2}^{T}(A_{t-1,t} - A_{0,t})$, measuring how much learning previous tasks improves performance on the next unseen task relative to the pretrained baseline. $\def\u#1{\underline{#1}}$ $\def\b#1{\mathbf{#1}}$
>
> |Method|Skill|Seg.|Assoc.|Plan.|Ant.V|Ant.N|
> |-|-|-|-|-|-|-|
> |FT|$\b{3.70}$|$19.89$|$9.85$|$0.09$|$2.83$|$1.57$|
> |ER|$2.93$|$18.53$|$0.33$|$0.81$|$\b{4.07}$|$\b{3.01}$|
> |DER++|$3.10$|$20.04$|$2.80$|$1.44$|$3.28$|$2.36$|
> |LwF|$2.64$|$20.55$|$\u{9.88}$|$0.99$|$2.61$|$2.32$|
> |SP+|$1.18$|$\u{22.98}$|$6.64$|$\b{2.14}$|$3.06$|$2.31$|
> |VISTA|$\u{3.11}$|$\b{27.68}$|$\b{11.66}$|$\u{1.57}$|$\u{3.61}$|$\u{2.48}$|
>
> > Training: Ego+Exo. "Seg.", segmentation; "Assoc.", association; "Ant.", anticipation; "V", verb; "N", noun; "FT", fine-tuning; "SP+", S-Prompt+.
>
> VISTA achieves the strongest FWT on segmentation, association, and is consistently competitive across other scenarios. This suggests that the subspace-routed adapter structure not only retains past knowledge but also provides a representation that benefits future task learning. We will include FWT in the revised evaluation.
>
> ### **Q2 & Limitation: CE4L and embodied decision-making problem.**
>
> CE4L is intended as an embodied perception** benchmark rather than an end-to-end decision-making benchmark. Its four tasks target perceptual capabilities that are important for embodied agents under viewpoint and task shifts, including **cross-view correspondence, temporal parsing, skill judgment, and future action prediction**. Although higher CE4L scores do not automatically guarantee higher downstream robotics success, these capabilities are clearly relevant to embodied decision making. To further connect CE4L to actual embodied tasks, we additionally evaluate VISTA on the continual VLA benchmark **LIBERO**[1], where it also performs strongly: it achieves the best **success-rate AUC** and the best **negative backward transfer** (NBT), showing that the same design is effective beyond CE4L. We will include these results and implementation details in the revised manuscript.
>
> |Method| AUC$(\uparrow)$|FWT$(\uparrow)$|NBT$(\downarrow)$|
> |-|-|-|-|
> |SeqFFT|$22.37_{\pm 0.27}$|$\u{76.13}_{\pm 0.97}$|$74.70_{\pm 1.05}$|
> |SeqLoRA|$21.37_{\pm 1.03}$|$73.10_{\pm 1.77}$|$71.64_{\pm 1.50}$|
> |PackNet|$4.84_{\pm 0.24}$|$37.20_{\pm 1.04}$|$41.34_{\pm 1.16}$|
> |ER|$\u{60.54}_{\pm 0.21}$|$\b{76.60}_{\pm 0.94}$|$\u{22.74}_{\pm 1.82}$|
> |VISTA|$\b{74.92}_{\pm 0.92}$|$74.17_{\pm 1.33}$|$\b{-0.39}_{\pm 1.17}$|
>
> > Continual LIBERO-10 performance. Model: DINOv2+CLIP-text+DiT flow matching, pretrained on LIBERO-90. Performance measured over 3 random runs.
>
> [1] LIBERO: Benchmarking Knowledge Transfer for Lifelong Robot Learning, NeurIPS2023

---

> > ### Author Rebuttal · Reviewer_RMdJ · 2026-04-01
> >
> > Thanks for the additional experiments and explanations. I have raised the rating.

---

> > > ### Author Response · Authors · 2026-04-01
> > >
> > > We are pleased that our response has fully addressed the concerns. We will incorporate these additions in the final version. Thank you once again for your support and valuable suggestions!

---

### Official Review · Reviewer_PwVH · 2026-03-11

**Soundness:** 3
**Presentation:** 3
**Significance:** 2
**Originality:** 3
**Overall Recommendation:** 4
**Confidence:** 4

**Summary:**

This paper addresses continual learning in egocentric-exocentric video understanding. The authors introduce a benchmark encompassing four tasks: skill assessment, action segmentation, cross-view retrieval, and action anticipation, evaluated across different viewpoint training and testing configurations. Then, they propose VISTA, a continual learning method that employs task-specific lightweight adapters with parameter isolation to prevent catastrophic forgetting. Experiments demonstrate that VISTA achieves near-joint-training performance with approximately zero forgetting across various tasks.

**Compliance With Llm Reviewing Policy:**

Affirmed.

**Final Justification:**

The author has supplemented the experiments and answered my concerns. The paper has reasonable contributions, so I give it a Weak accept

**Key Questions For Authors:**

The VISTA method is a relatively general multi-task continuous learning approach. Has it been compared with other general continuous learning datasets?

Other questions are the same as the weaknesses

**Limitations:**

The primary challenge in cross-view understanding lies in understanding physical space rather than semantic understanding, such as relative positions of objects in space and directions of motion. However, in evaluation settings such as action planning, spatial understanding is not involved and there is no related discussion.

**Strengths And Weaknesses:**

Strengths:

1.The paper constructs a cross-view continual learning benchmark.

2.The paper designs VISTA, a task-level continual learning model that learns independent lightweight adapters for each task, avoiding inter-task interference through parameter isolation. During inference, adapters are selected through representations, and the router requires no training, demonstrating good cross-task capability retention.

3.Extensive cross-experiments between viewpoints and tasks validate the model's effectiveness.

Weaknesses:

1. The primary challenge in cross-view understanding lies in understanding physical space rather than semantic understanding, such as relative positions of objects in space and directions of motion. However, in evaluation settings such as action planning, spatial understanding is not involved and there is no related discussion.

2. The benchmark design is somewhat disconnected from VISTA. The benchmark focuses on cross-view understanding, while VISTA lacks motivation-driven thinking and innovation specifically targeting this aspect. In my view, VISTA can independently serve as a multi-task continual learning method, making the paper's theme unclear.

3. VISTA requires different adapters for different tasks, resulting in linear growth of adapters with the number of tasks. In embodied scenarios where tasks can sometimes be numerous, potentially reaching 1000+, what would be the impact on the method?

---

> ### Author Rebuttal · Authors · 2026-03-31
>
> We thank you for the constructive comments and provide point-by-point responses to **Weaknesses (W)** and **Questions (Q)** below. Revisions will be added in the final version.
>
> ### **W1 & limitation: Spatial reasoning.**
>
> Spatial reasoning is indeed important for cross-view understanding, and recent work has made progress on 3D pose estimation, object correspondence, and spatial grounding across viewpoints [1,2,3]. Semantic cross-view understanding is also a promising direction. Learning view-invariant representations for action recognition, skill assessment, and temporal reasoning across ego and exo views has received growing attention [1,4,5], reflecting that understanding what is happening and how well also matters in cross-view settings. We agree that in embodied interactive environments, planning requires fine-grained spatial groundings. CE4L's action planning predicts future procedural steps from video, and its cross-view challenge lies in transferring this ability across the ego-exo appearance gap. We will clarify this scope distinction and add the relevant discussion in the revised manuscript.
>
> [1] Ego-Exo4D: Understanding skilled human activity from first- and third-person perspectives, CVPR2024
>
> [2] 3DSRBench: A comprehensive 3D spatial reasoning benchmark, ICCV2025
>
> [3] Thinking in space: How multimodal large language models see, remember and recall spaces, CVPR2025
>
> [4] Learning fine-grained view-invariant representations from unpaired ego-exo videos via temporal alignment, NeurIPS2023
>
> [5] Viewpoint rosetta stone: unlocking unpaired ego-exo videos for view-invariant representation learning, CVPR2025
>
> ### **W2: Benchmark-method alignment.**
>
> CE4L exposes a CL difficulty that intensifies under cross-view shift: task identity inference becomes harder when ego and exo features of the same task occupy different regions of the representation space. A separately trained task classifier can itself forget or become view-biased as new tasks arrive. VISTA addresses this by avoiding any learned classifier. Its whitened subspace router normalizes viewpoint-dependent variance through whitening and captures task-specific variation structure through subspace fitting, making routing more robust to the ego-exo distributional gap than cosine or K-NN alternatives (Fig. 3(d)). Therefore, although the design principle of VISTA is broadly useful, the multi-view setting is where its advantage is most pronounced and well-aligned. We will make this motivation clearer.
>
> ### **W3: Scalability.**
>
> VISTA does introduce one adapter per task, so its storage grows linearly with task count. However, the per-task overhead is small. In the largest current setting (skill assessment, $C=d=1024$, $r=64$, $q=32$), this is only about **0.17M scalars per task** ($\approx 0.68$ MB in fp32), which remains far smaller than replay-based storage (e.g., **56.20M** scalars for ER on skill assessment and **148.73M** on anticipation). This makes VISTA practical for the CE4L regime and, more broadly, for many realistic continual streams where memory efficiency matters. We will include this scaling analysis in the revised manuscript and discuss techniques like adapter merging, compression, and hierarchical routing as directions for very long task streams.
>
> ### **Q1: VISTA on general CL.**
>
> VISTA is general at the architectural level, although our main motivation is to provide a strong replay-free baseline for continual multi-view video learning under CE4L. To verify its broader applicability, we additionally evaluate a variant, **VISTA-Prompt**, by replacing adapters with prompts on standard general CL benchmarks under a more realistic **online, blurry-boundary** setting [1,2], which is more similar to the video stream in CE4L, rather than the overly idealized offline image classification setup. The results show that the same routing strategy transfers well beyond CE4L: VISTA-Prompt achieves the best overall performance on **CIFAR-100** and **ImageNet-R**, and remains competitive on **CUB-200**. For embodied tasks, see our response to Reviewer RMdJ (W4). We will include these results and implementation details in the revised manuscript.
>
> ||CIFAR-100||ImageNet-R||CUB-200||
> |-|-|-|-|-|-|-|
> ||$A_{\rm{AUC}}$|$A_{\rm{last}}$|$A_{\rm{AUC}}$|$A_{\rm{last}}$|$A_{\rm{AUC}}$|$A_{\rm{last}}$|
> |L2P|76.23|79.11|44.40|42.03|64.30|61.42|
> |DualPrompt|76.04|76.62|46.13|40.80|65.03|62.43|
> |CODA-P|79.13|80.91|51.87|48.09|66.01|62.90|
> |MISA|80.35|80.75|51.52|45.08|65.40|60.20|
> |HiDe-Prompt|77.10|81.77|53.77|49.87|**67.05**|**67.12**|
> |VISTA-Prompt|**81.10**|**83.52**|**54.50**|**52.47**|65.03|64.72|
>
> > 5 CL tasks per dataset. $A_{\rm{AUC}}$: anytime accuracy AUC. $A_{\rm{last}}$: final average accuracy. Model: ViT-B/16 (ImageNet-21K). 5 random runs.
>
> [1] Online class incremental learning on stochastic blurry task boundary via mask and visual prompt tuning, ICCV2023.
>
> [2] Advancing prompt-based methods for replay-independent general continual learning, ICLR2025

---

> > ### Author Rebuttal · Reviewer_PwVH · 2026-04-03
> >
> > Thanks for the rebuttal; it addresses some of my concerns. However, the key issue still remains: the motivation–method alignment is unclear. Specifically, is VISTA meant to be a general continual learning method, or a multi-view (ego–exo) spatial understanding method? At the moment, VISTA seems largely task-incremental and generic, and the paper does not clearly explain how its design is directly driven by the main difficulties in multi-view understanding (especially beyond semantic/domain shift, e.g., spatial/physical reasoning).
> >
> > Thanks for the reply rebuttal. I have raised the rating.

---

> > > ### Author Response · Authors · 2026-04-03
> > >
> > > We appreciate your thoughtful follow-up and would like to address the positioning question carefully.
> > >
> > > Spatial and physical reasoning is an important perspective on cross-view understanding, and we agree that geometric correspondence and physical relationships across views are key challenges. At the same time, **semantic cross-view understanding**, where models learn consistent representations of the same events, actions, and skills regardless of viewpoint, is also a fundamental objective of multi-view video learning. Spatial reasoning addresses how objects and scenes relate geometrically across views, while semantic understanding addresses whether high-level knowledge (action categories, temporal structure, skill quality) transfers across the appearance gap between viewpoints. These two perspectives are complementary, and combining them is an exciting future direction.
> > >
> > > CE4L approaches video continual learning from the semantic level because semantic understanding underpins both **within-view retention and cross-view generalization**: a continual learner must preserve and transfer knowledge of actions, temporal structure, and skills whether evaluated within or across viewpoints. Prior video CL benchmarks (vCLIMB, ViLCo) test only single-view semantic retention. CE4L extends this by additionally evaluating **cross-view generalization under multi-view training**, providing a more comprehensive testbed for video CL research. This richer evaluation reveals a failure pattern that single-view benchmarks cannot expose: within each scenario (segmentation, skill, association, anticipation), models trained sequentially on mixed ego-exo data can develop **view-specific representations** that retain within-view performance but fail to transfer across views (Figs. 3-5). Our analysis traces this failure to task routing: under mixed viewpoints, standard routers (cosine, K-NN) that perform well in single-view CL degrade substantially because each procedural activity's features are spread across the ego-exo distributional gap rather than tightly clustered (Fig. 3(d)).
> > >
> > > VISTA's design follows directly from this observation. When ego and exo data are mixed, each task's features reflect both task semantics and viewpoint statistics. Standard routers (cosine, K-NN) only see where each task's features are centered on average. This signal weakens when ego and exo features of the same task spread in different directions, and in CL the router cannot be retrained as new tasks arrive. VISTA addresses this at two levels. **Whitening** normalizes per-dimension variance, down-weighting dimensions where viewpoint variation dominates so that task prototypes become more comparable across views. The **subspace residual** then captures not just the center but the shape of each task's feature cloud via principal components: two tasks can share similar centers but differ in how their features vary, and the residual score tells them apart by how well a sample aligns with each task's principal directions. This matches our finding that the whitened-subspace router gains most on segmentation, where task means are close but temporal variation structure differs across procedures. The soft top-$K$ mixture further allows interpolation near task boundaries rather than hard selection. The routing ablation confirms this design across all five CE4L benchmarks, with the largest gains on scenarios with the most viewpoint mixing (segmentation, anticipation).
> > >
> > > We will state the scope boundary with respect to spatial reasoning explicitly and revise the motivation to make this observation-to-design connection (routing failure under mixed views → VISTA's whitened-subspace routing) clearer from the start. We hope you find these clarifications sufficient to reconsider the score. If there are further questions, please let us know and we are happy to discuss.

---

### Official Review · Reviewer_awR6 · 2026-03-12

**Soundness:** 3
**Presentation:** 2
**Significance:** 3
**Originality:** 3
**Overall Recommendation:** 4
**Confidence:** 5

**Summary:**

Paper considers a benchmark for continual learning from ego-centric and exo-centric video streams, which naturally covers viewpoint distribution shifts and prevents learning viewpoint shortcuts to solve sequential tasks.  An adatpter-based method is proposed that utilizes task subspaces and routing. Evaluation on four relevant benchmark tasks demonstrate classic CL methods suffer from viewpoint distribution shifts, while the proposed VISTA method has favorable performance.

**Compliance With Llm Reviewing Policy:**

Affirmed.

**Final Justification:**

I am satisfied with the rebuttal and interaction, upgrading my recommendation accordingly. I expect the authors to incorporate all improvements in the updated main manuscript.

**Key Questions For Authors:**

Please address all the weaknesses as indicated above.

**Limitations:**

The paper contains no limitation section. This is another weakness.

**Strengths And Weaknesses:**

**Strong**

Good problem statement, and interesting benchmark that makes a lot of sense. Paper is also well written in general. Extensive set of experiments substantiate the claims.


**Weak**

* The benchmark builds heavily on EgoExoLearn by Huang et al., 2024. From a negative viewpoint the proposed benchmark feels a bit as a bake-off by 'just' adding a continual variant. The contribution over Huang et al is not well described and motivated.

* The related work section is moved to the supplemental. This is unacceptable. Works needs proper embedding in relevant literature. Also several relevant works on video generalization and robustness benchmarking are not discussed at all, see for example:
   * S Grover et al. Revealing the unseen: Benchmarking video action recognition under occlusion. NeurIPS 2023.
   * SK Maharana et al. 𝙰𝚅𝚁𝙾𝙱𝚄𝚂𝚃𝙱𝙴𝙽𝙲𝙷: Benchmarking the Robustness of Audio-Visual Recognition Models at Test-Time. NeurIPS 2025
   * MC Schiappa et al. A large-scale robustness analysis of video action recognition models. CVPR 2023.
   * FM Thoker et al. How severe is benchmark-sensitivity in video self-supervised learning? ECCV 2022.
   * S Vyas et al. Multi-view Action Recognition Using Cross-View Video Prediction. ECCV 2020
   * Y Zhang et al. Audio-Adaptive Activity Recognition Across Video Domains. CVPR 2022.

* The baseline encoders used are quite outdated (e.g. relying on I3D/RAAN  and CLIP/TA3N). This makes the benchmark not very current.

* The VISTA method has a selection mechanism baked in, which seems to offer an advantage for mixed-viewpoints settings. What alternative routing mechanisms were considered, could this be done in a test-time adaptation fashion? I would also like to see more ablations on the routing mechanism, what is the impact of the mixture weights, how sensitive is the approach? Why is the skill task so easy (figure 3d), why is segmentation profiting so much from the whitened subspace router?

* It is not clear from the paper what will be publicly released as part of the proposed benchmark and method.

* In terms of presentation the work over-emphasizes tasks and empirical results, I would have preferred more theoretical motivation, a better embedding and more clear contribution claim, also in light of overall novelty (benchmark builds on Huang et al, backbones are outdated, etc)

---

> ### Author Rebuttal · Authors · 2026-03-31
>
> We sincerely thank the reviewer and provide responses below.
>
> ### **W1,W6: Novelty & motivation**
>
> CE4L targets an unaddressed gap in video CL: the coupled shift of tasks and viewpoints. As in CL benchmarks, the contribution lies in the protocol and insights it enables.
>
> First, CE4L contributes a continual multi-view protocol with four video tasks, three regimes, within- and cross-view evaluation, and unified metrics ($\bar{A}_T$, $\bar{F}_T$, anytime curves), explicitly separating **knowledge retention from cross-view generalization**. Second, it reveals a failure mode prior video CL benchmarks cannot expose: within-view forgetting can decrease while cross-view performance still degrades (Figs. 3-5); in segmentation, the gap persists even under joint training, indicating **retention and transferability are distinct failure modes** and that the limitation is partly representational rather than only CL-specific. Third, it adapts 8 CL methods to 4 tasks, providing a **reusable research platform** beyond EgoExoLearn.
>
> For motivation, mixed ego/exo data shifts both feature centers and variation. Mean-based routers capture only centers; whitening suppresses viewpoint-dominant variance; the subspace residual captures task-specific variation directions, explaining VISTA's advantage when tasks have nearby means but different structure (see W4). We will revise to separate CE4L, VISTA, and empirical findings.
>
> ### **W2: Related work**
>
> We will cite the suggested references and move related work to the main text. These works study video robustness under occlusion, multimodal corruptions, benchmark sensitivity, cross-modal adaptation, and cross-view prediction. The key distinction is the axis: they mostly test fixed models after joint training, whereas CE4L studies sequential adaptation across heterogeneous tasks, with viewpoint shift persisting during training and interacting with task interference. Thus the settings are complementary.
>
> ### **W3: Backbone**
>
> The backbones follow EgoExoLearn for controlled comparison; CE4L is not tied to them. Replacing with **EgoVLPv2** preserves the relative ranking and keeps VISTA strongest, confirming findings are not backbone artifacts. Full EgoVLPv2 and VideoMAEv2 (CVPR 2023) results will follow in discussion.
>
> ||Skill|Assoc.||Plan.||Ant.||||
> |-|-|-|-|-|-|-|-|-|-|
> |||E→X|X→E|Ego|Exo|Ego-V|Ego-N|Exo-V|Exo-N|
> |ER|76.92|29.30|23.40|81.43|82.21|21.72|26.90|21.83|23.81|
> |DER++|77.08|32.20|28.70|**79.50**|79.97|22.46|**29.95**|22.63|25.29|
> |LwF|69.05|32.40|30.10|80.78|81.66|18.71|26.30|18.96|22.63|
> |S-Prompt+|77.36|32.60|28.40|81.81|81.15|19.86|22.94|19.77|20.52|
> |VISTA|**78.18**|**33.60**|**30.70**|79.72|**79.42**|**23.72**|28.12|**23.71**|**25.94**|
>
> ### **W4: Routing**
>
> We additionaly compare Mahalanobis (per-task mean+diagonal variance) and GMM (q=32) with Fig. 3(d):
>
> |Router|Skill|Seg.|Assoc.|Plan.|Ant.|
> |-|-|-|-|-|-|
> |K-NN|88.50|55.95|57.96|44.56|54.94|
> |Cosine|97.58|56.14|62.25|42.78|60.55|
> |Wh. Cosine|98.31|57.14|**70.96**|42.78|61.15|
> |GMM|88.06|64.29|59.25|44.81|54.23|
> |Mahalanobis|88.78|65.00|62.88|50.63|53.58|
> |Wh. Subspace|**98.86**|**75.00**|63.42|**58.48**|**73.04**|
>
> First-order methods are strong on skill but degrade on longer inputs. GMM/Mahalanobis help segmentation via density modeling but remain below the whitened-subspace router. On association, whitened cosine leads due to short clips and contrastive training. The whitened-subspace router is most consistent.
>
> We also compare top-1, top-2 average, and residual-weighted top-2 mixture:
>
> ||Skill|Seg.||Assoc.||Plan.||Ant.||||
> |-|-|-|-|-|-|-|-|-|-|-|-|
> |||Ego|Exo|E→X|X→E|Ego|Exo|Ego-V|Ego-N|Exo-V|Exo-N|
> |Oracle|80.42|47.17|33.23|45.07|49.87|82.35|81.04|35.35|22.57|35.45|23.70|
> |Top-1|79.62|42.70|27.74|41.50|38.10|83.85|82.72|34.68|20.81|32.37|20.80|
> |Top-2 Avg.|76.05|42.15|30.91|40.50|37.55|83.96|81.14|32.82|15.10|31.13|15.30|
> |Top-2 Mix.|**80.38**|**45.35**|**31.41**|**44.74**|**38.29**|**82.81**|**80.74**|**35.23**|**25.23**|**32.77**|**26.01**|
>
> For **skill**, short clips yield well-separated geometry (Fig. 5), so top-1 suffices. On harder benchmarks the residual-weighted top-2 outperforms both alternatives, showing confidence-aware weighting matters. For **segmentation**, shared temporal patterns across long procedural videos make prototype matching insufficient; modeling principal variation directions explains the whitened subspace router's largest gain.
>
> Regarding **TTA**: CE4L targets generalization acquired during training under single-pass inference; TTA adapts at deployment using test batches, complementary but inference-heavier than VISTA router. Combining CL with TTA is a promising direction.
>
> ### **W5,W7: Release & limits**
>
> The supplementary includes our full codebase (CE4L framework, all methods including VISTA); we will open-source upon acceptance.
>
> Limitations: CE4L uses one ego-exo corpus and a task-incremental setting. Extending to more datasets and task-agnostic or online CL is a natural direction.

---

> > ### Author Rebuttal · Reviewer_awR6 · 2026-04-03
> >
> > I am mostly satisfied with the rebuttal, several of my concerns are well addressed. The only remaining point is that it is still insufficiently clear what this paper adds over the EgoExoLearn benchmark by Huang et al., 2024. I would prefer a more explicit description of the difference.

---

> > > ### Author Response · Authors · 2026-04-03
> > >
> > > We thank you for the positive update. We are glad that most concerns have been well addressed and are happy to clarify the last remaining point below.
> > >
> > > EgoExoLearn provides a rich multi-view video dataset along with static benchmarks that train models on ego, exo, or combined data and evaluate cross-view transfer at convergence. CE4L introduces a **different evaluation paradigm** on top of the same data and training configurations. It poses a question that static evaluation cannot address: when tasks arrive sequentially from mixed ego-exo streams, does **cross-view transfer degrade independently of within-view retention**, and how does this interaction evolve over the learning trajectory?
> > >
> > > CE4L provides a complete CL benchmark infrastructure to answer this question. The table below summarizes what is shared with EgoExoLearn and what CE4L contributes. All CE4L-specific components (right column) are included in the supplementary materials and will be open-sourced upon acceptance.
> > >
> > > ||EgoExoLearn|CE4L (ours)|
> > > |-|-|-|
> > > |Multi-view video data & annotations|✓|✓ (shared)|
> > > |Task-specific evaluation metrics|✓|✓ (shared)|
> > > |Ego / Exo / Ego+Exo training|✓|✓ (shared)|
> > > |Sequential task streams||✓|
> > > |CL metrics ($\bar{A}_T$, $\bar{F}_T$, FWT, curves)||✓|
> > > |Per-step within-view & cross-view evaluation||✓|
> > > |8 CL baselines adapted to video||✓|
> > > |VISTA method||✓|
> > >
> > > Our experiments then confirm the question CE4L poses. Figs. 3-5 show that within-view forgetting can decrease while cross-view performance continues to degrade, revealing that **retention and cross-view transferability are distinct failure modes**. In segmentation, the cross-view gap persists even under joint training, pointing to a representational limitation independent of the CL protocol. These observations are only visible through sequential evaluation with per-step cross-view metrics. Since CE4L's contribution is at the evaluation paradigm level, the protocol and metrics are not tied to a specific dataset or backbone and can naturally extend to other multi-view video corpora as they become available. As stated in W3, we now provide the results with both EgoVLPv2 and VideoMAEv2 backbones:
> > >
> > > **EgoVLPv2 (ICCV 2023):**
> > >
> > > |||Skill|Assoc.||Plan.||Ant.||||
> > > |-|-|-|-|-|-|-|-|-|-|-|
> > > ||||E→X|X→E|Ego|Exo|Ego-V|Ego-N|Exo-V|Exo-N|
> > > ||FT|65.08|26.20|20.30|82.91|84.71|16.47|20.81|16.62|18.61|
> > > ||ER|76.92|29.30|23.40|81.43|82.21|21.72|26.90|21.83|23.81|
> > > ||DER++|77.08|32.20|28.70|**79.50**|79.97|22.46|**29.95**|22.63|25.29|
> > > ||EWC|70.20|31.20|29.00|82.54|82.26|17.62|21.65|17.85|19.23|
> > > ||LwF|69.05|32.40|30.10|80.78|81.66|18.71|26.30|18.96|22.63|
> > > ||L2P+|76.66|31.00|29.10|81.70|81.89|19.22|23.21|19.34|20.22|
> > > ||S-Prompt+|77.36|32.60|28.40|81.81|81.15|19.86|22.94|19.77|20.52|
> > > ||VISTA|**78.18**|**33.60**|**30.70**|79.72|**79.42**|**23.72**|28.12|**23.71**|**25.94**|
> > >
> > > **VideoMAEv2 (CVPR 2023):**
> > >
> > > |||Skill|Plan.||Ant.||||
> > > |-|-|-|-|-|-|-|-|-|
> > > ||||Ego|Exo|Ego-V|Ego-N|Exo-V|Exo-N|
> > > ||FT|71.15|81.59|82.04|20.38|26.24|20.35|22.69|
> > > ||ER|77.66|78.55|79.64|26.34|31.38|27.20|26.81|
> > > ||DER++|78.46|**77.72**|78.64|26.78|31.26|27.67|26.82|
> > > ||EWC|76.02|80.79|80.50|21.26|28.18|22.29|24.07|
> > > ||LwF|77.96|78.54|81.18|23.05|31.76|23.02|**28.33**|
> > > ||L2P+|79.88|78.90|79.72|22.49|28.59|23.72|24.30|
> > > ||S-Prompt+|80.01|79.61|80.77|23.39|28.37|23.31|25.00|
> > > ||VISTA|**81.67**|78.36|**78.31**|**27.87**|**32.12**|**28.84**|27.83|
> > >
> > > > Segmentation and VideoMAEv2 association experiments are ongoing. Best in **bold**.
> > >
> > > The relative ranking of CL methods and VISTA's consistent advantage are preserved across both backbones, confirming that our findings are transferable across encoder choices. We will include the complete results and a more explicit discussion of the CE4L vs. EgoExoLearn distinction in the final version. We hope you regard these clarifications as sufficient to reconsider the score. Please let us know if any further questions remain.

---

### Official Review · Reviewer_S2nj · 2026-03-13

**Soundness:** 3
**Presentation:** 4
**Significance:** 4
**Originality:** 4
**Overall Recommendation:** 4
**Confidence:** 3

**Summary:**

The authors of this work propose a new benchmark (CE4L) for continual learning in embodied agents that captures real-world heterogeneous video tasks (ego, exo, and mixed view), and further introduce (VISTA), a parameter-efficient method that stores task-specific updates in lightweight adapters and routes them using residual distance to whitened subspaces estimated from second-order statistics.

**Compliance With Llm Reviewing Policy:**

Affirmed.

**Final Justification:**

The authors have addressed most of my concerns within their rebuttal. As such, I shall keep my current rating.

**Key Questions For Authors:**

1.0 The authors should ablate different subspace ranks (q), to assess optimal rank (q) for routing accuracy and final performance. The authors should further justify why q has been given a rank of 32.

2.0 The authors may further look to investigate how VISTA performs whilst dealing with out-of-distribution inputs from unseen tasks. This would further strengthen the task generalisation capabilities of the proposed method.

3.0 The authors may consider highlighting the number of total tasks across all CE4L scenarios. In addition, please further consider reporting the train:test split, frame counts, FPS and resolutions within CE4L.

**Limitations:**

Yes

**Strengths And Weaknesses:**

Soundness:
This papers showcases thorough extensive experimental results. The experiments are well-designed on their introduced benchmark (CE4L), and their (VISTA) method showcases strong performances across tasks.

Tables 2, 3, 4, 5, and 6 show that the authors proposed VISTA method is a strong method across multiple tasks against SOTA methods.
Table 7 showcases a ablation study of the component combinations within VISTA on CE4L (across tasks).
Table 8 shows an evaluation of efficiency on continual cross-view association. VISTA has comparable Trainable(M) and Total(M) parameter counts compared to SOTA, whilst having second best storage [marginal] and 2nd best Time (ms) [marginal].

Presentation:
This paper is well presented, and well written and well structured. There are no major presentation issues to note.
It highlights limitations of prior literature well and discusses its differences through contributions driven from such limitations.

Significance and Originality:
The authors propose a unified multiview CL benchmark benchmark (CE4L) that spans heterogeneous video tasks (ego, exo, and mixed view) to address real-world coupled shifts in continual learning.
The authors propose VISTA is an efficient continual learning method for multi-view video tasks. It encodes task knowledge in lightweight adapters with a training-free subspace routing mechanism.

---

> ### Author Rebuttal · Authors · 2026-03-31
>
> We sincerely thank you for your constructive comments. Below we provide point-to-point responses to **Weakness (W)** and **Questions (Q)** and will add corresponding revisions in the final version.
>
> ### **Q1: Subspace rank $q$.**
>
> Fig. 3(c) already reports sensitivity to $q$ on continual anticipation under Ego+Exo training, where performance remains stable in a moderate range. We further ablated $q \in \{4,8,16,32,64\}$ and measured **routing accuracy** and **final continual performance** on skill assessment, anticipation, and planning.
>
> |$q$|Skill||Ant.||Plan.||
> |-|-|-|-|-|-|-|
> ||Routing|$\bar{A}_T\uparrow$|Routing|$\bar{A}_T\uparrow$|Routing|ED@8$\downarrow$|
> |4|98.53|79.59|55.70|33.16|51.90|84.09|
> |8|98.35|79.55|63.34|34.39|50.89|83.73|
> |16|98.21|79.49|68.55|34.77|54.43|83.57|
> |32|**98.86**|**80.38**|73.04|35.23|**58.50**|**82.81**|
> |64|97.25|79.23|**74.98**|**35.86**|53.92|83.94|
>
> > Training: Ego+Exo. Test: Ego. "Ant.", anticipation; "Plan.", planning.
>
> Small ranks ($q=4,8$) are slightly under-expressive, while $q=64$ may improve in one setting but lacks consistency. In contrast, **$q=32$ provides the best overall tradeoff**, achieving the strongest or near-strongest routing and downstream performance across tasks. We therefore use $q=32$ as a **shared default**.
>
> ### **Q2: Generalization on unseen-task / OOD inputs.**
>
> In the current CE4L protocol, evaluation is performed on the union of seen tasks, and the router selects only among previously learned adapters. To probe generalization beyond this standard setting, we additionally evaluate **future unseen tasks during the continual process** using $\tilde{A}=\frac{1}{T-1}\sum_{i=1}^{T-1}\left(\frac{1}{T-i}\sum_{j=i+1}^T A_{i,j}\right)$, the upper triangle mean of matrix $\\{A_{i,j}\\}$, where $A_{i,j}$ is the performance on task $j$ after learning task $i$.
>
> |Method|Skill|Seg.|Assoc.|Plan.|Ant.V|Ant.N|
> |-|-|-|-|-|-|-|
> |FT|47.69|18.95|30.09|88.55|22.48|12.50|
> |ER|47.37|18.11|28.82|87.69|24.29|14.90|
> |DER++|47.51|18.84|30.70|87.93|26.10|14.85|
> |LwF|46.43|20.02|31.09|87.52|24.94|14.96|
> |SP+|44.61|19.92|26.21|87.97|20.97|12.45|
> |VISTA|**48.04**|**24.53**|**31.80**|**87.18**|**26.89**|**15.04**|
>
> > Training: Ego+Exo. Test: Ego+Exo. "Seg.", segmentation; "Assoc.", association; "V" or "N", anticipation on verb or noun. "FT", fine-tuning; "SP+", S-Prompt+.
>
> Across CE4L under Ego+Exo training, VISTA shows the strongest unseen-task generalization, indicating that subspace-based routing remains robust even for inputs from future tasks outside the seen set.
>
> We further evaluate **tasks excluded from the original training-evaluation pipeline** of CE4L for segmentation/association/planning (as described in Q3). Although these tasks are never used in the continual training stream, VISTA achieves the strongest overall performance in most cases.
>
> ||Seg.||Assoc.||Plan.||
> |-|-|-|-|-|-|-|
> ||Ego|Exo|E→X|X→E|Ego|Exo|
> |FT|46.60|36.58|6.00|14.00|84.00|82.96|
> |ER|42.54|38.28|6.00|17.00|84.11|79.00|
> |DER++|47.92|40.70|10.00|15.00|84.55|78.15|
> |LwF|41.29|41.18|8.00|13.00|**83.45**|78.10|
> |SP+|47.54|38.13|**17.00**|17.00|84.01|80.14|
> |VISTA|**48.36**|**42.58**|**17.00**|**19.00**|83.97|**77.14**|
>
> > Training: Ego+Exo; Test: EgoExoLearn data not used in current CE4L pipeline. "E→X", Ego→Exo; "X→E", Exo→Ego.
>
> These results confirm that VISTA generalizes robustly beyond standard seen-task CE4L to unseen-task / OOD inputs.
>
> ### **Q3: Benchmark statistics.**
>
> We summarize them in Table 1 and Appendix. We will add a dedicated table: CE4L is built on EgoExoLearn, which contains **8 high-level procedural tasks** and **745 videos**. On top of this, CE4L instantiates continual streams with **4 tasks** for skill assessment, **4** for temporal action segmentation, **5** for cross-view association, **8** for action anticipation, and **4** for action planning. The default segmentation / association / planning streams use fewer than 8 tasks because the three lab procedures (tasks 6–8) are much sparser and lack annotations in some splits, and kitchen task 2 has no ego-train videos for segmentation/planning. We will also report the **train/val/test splits, frame counts, FPS, and input resolutions** for each scenario in the revised paper. A compact version is as follows.
>
> |Scenario|Tasks|Ego split|Exo split|Input|
> |-|-|-|-|-|
> |Skill|4 action groups|13,722 / 10,518 pairs|—|10 segments per clip|
> |Segmentation|4 procedural tasks|173 / 28 / 55 vids|176 / 21 / 26 vids|Full video, 5× downsampled|
> |Association|5 procedural tasks|22.7k clips (314 vids)|5.4k clips (249 vids)|4 frames per clip|
> |Anticipation|8 procedural tasks|34.6k / 7.7k / 17.3k clips|6.1k / 2.1k / 4.7k clips|2 s context, 5 frames at 5 fps|
> |Planning|4 procedural tasks|2,171 / 395 / 734 clips|1,847 / 205 / 270 clips|2 s context, 5 frames at 5 fps|
>
> > Split format: train / val / test. Skill uses train / val only. Association evaluation: 800 val / 2,000 test direction-balanced MCQs. All videos are standardized to 25 fps, 224×224.

---

> > ### Author Rebuttal · Reviewer_S2nj · 2026-04-03
> >
> > Thank you to the authors for their thorough and insightful rebuttal. I have read through the authors rebuttal carefully and they have addressed each of my concerns. I recommend that the authors include the subspace q ranking ablation within their revised version. I shall keep my current rating.

---

> > > ### Author Response · Authors · 2026-04-04
> > >
> > > Thank you very much for your valuable feedback. We will ensure that all discussed modifications and additional content are carefully incorporated into the final version. Thanks again for your time and constructive suggestions to improve our work.

---

### Decision · Program_Chairs · 2026-04-30

**Decision:**

Accept (regular)

**Comment:**

This work propose a new benchmark for continual learning in embodied agents that captures real-world heterogeneous video tasks (ego, exo, and mixed view), and further introduce (VISTA), a parameter-efficient method that stores task-specific updates in lightweight adapters and routes them using residual distance to whitened subspaces estimated from second-order statistics. All reviewers are positive to this submission after the rebuttal.